# SCOUT: Active Information Foraging for Long-Text Understanding with Decoupled Epistemic States

Zhenliang Zhang [1 2 †]   Wenqing Wang [1 2 †]   Yong Hu [2]   Yaming Yang [2]
Jiaheng Gao [2]   Chen Shen [2]   Xiaojun Wan [1]

## Abstract

Long-Text Understanding (LTU) at million-token scale requires balancing reasoning fidelity with computational efficiency. Frontier long-context LLMs can process millions of token contexts end-to-end, but they suffer from high token consumption and attention dilution. In parallel, specialized LTU agents often sacrifice fidelity through task-agnostic abstractions like graph construction or indexing. We identify a key insight for LTU: query-relevant information is typically sparse relative to the full document, so effective reasoning should rely on a query-sufficient subset rather than the entire context. To address this, we propose SCOUT, a new paradigm for LTU that **shifts from passive processing to active information foraging**. It treats the document as an explorable environment and answers from a compact, provenance-grounded epistemic state. Guided by state-level gap diagnosis, SCOUT adaptively alternates between coarse-to-fine exploration and anchored state updates that progressively contract its epistemic state toward query sufficiency. Experiments show that SCOUT matches state-of-the-art proprietary models while reducing token consumption by up to $8\times$. Moreover, SCOUT remains stable as context length scales, substantially alleviating the practical cost–performance trade-off. Resources are available at our Project Page.

## 1. Introduction

Long-Text Understanding (LTU) is a major AI frontier, yet it remains challenging because useful information is often sparse and scattered across long documents, surrounded by

[†]Work done during internships at WeChat, Tencent. [1]Peking University [2]WeChat, Tencent. Correspondence to: Xiaojun Wan <wanxiaojun@pku.edu.cn>.

*Proceedings of the $43^{rd}$ International Conference on Machine Learning*, Seoul, South Korea. PMLR 306, 2026. Copyright 2026 by the author(s).

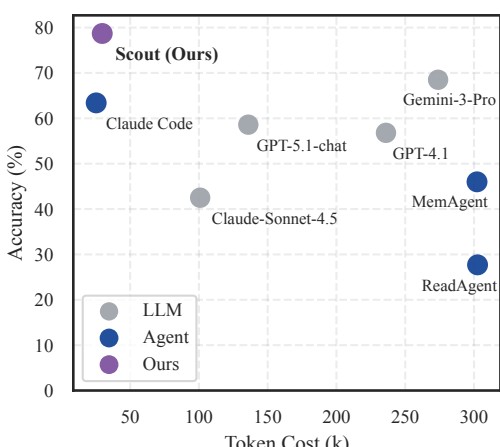

*Figure 1.* **Accuracy–cost trade-off.** Overall accuracy vs. token cost (k); SCOUT offers the best trade-off.

substantial irrelevant content.

Current approaches generally fall into three paradigms. First, native long-context LLMs (e.g., Gemini-3-Pro (Google DeepMind, 2025), GPT-5 (OpenAI, 2025b)) ingest massive contexts directly, but this brute-force strategy incurs high inference cost and suffers attention dilution at scale. Second, retrieval-augmented generation (RAG) reduces cost by retrieving a small subset of passages (Lewis et al., 2020; Izacard & Grave, 2021), yet chunk-level retrieval can weaken global coherence and long-range dependency reasoning, especially on multi-hop aggregation tasks. Third, specialized LTU agents aim to improve efficiency via decomposition and navigation (Yu et al., 2025a; Lee et al., 2024; Li et al., 2024b), but they often rely on task-agnostic pre-processing or entangle exploration traces with reasoning, compromising fidelity and robustness for long-range dependency reasoning.

As the frontier moves from Needle-in-a-Haystack retrieval to complex reasoning over million-token sequences, the trade-offs implicit in these paradigms become increasingly binding. We identify this as the **LTU Trilemma**, where systems struggle to simultaneously satisfy: *(1) extreme scalability* to handle ultra-long streams; *(2) high information*

*fidelity* to preserve nuance without lossy compression; and *(3) inference efficiency* to maintain low token cost.[1]

We attribute this bottleneck to a **Task-Agnostic Processing Trap**. Existing paradigms often pre-process long documents (via indexing, compression, or encoding) without conditioning on the downstream query. Concretely, building indices/graphs/gists *before* the query forces the system to commit to a fixed abstraction: it spends budget constructing and traversing structure over largely irrelevant regions, while any detail omitted at construction time becomes unrecoverable for downstream reasoning. As a result, computation is misallocated and query-critical, position-anchored cues can be lost, hurting both efficiency and fidelity. This observation motivates an **information sparsity assumption**: for a given query, the information needed to answer is typically a small, query-sufficient subset of the full document. Efficient LTU should therefore focus on discovering and consolidating this subset into a compact reasoning substrate, rather than committing computation to query-agnostic processing.

To exploit this sparsity, we transition from **passive text reception to active information foraging**. Building on this principle, a long document is not a static sequence to be processed end-to-end, but an environment to be explored on demand. We introduce **SCOUT** (**S**trategic **C**ontext **O**bservation for **U**nderstanding **T**ext), a new paradigm that enables agents to navigate raw text directly via interactive, goal-directed exploration, preserving raw-text fidelity while avoiding chunking, indexing, and embedding.

To make such interactive exploration reliable for long-range reasoning, SCOUT introduces a strict separation between *where the agent explores* and *what it has established as query-relevant knowledge* for downstream reasoning. Specifically, it maintains a compact, provenance-grounded Epistemic State as the sole substrate for answering, while the full navigation trace is used only to guide further exploration. This decoupled design prevents the growing interaction history from becoming a noisy reasoning context, and ensures that each committed unit is verifiable via its provenance anchor. In this sense, SCOUT retains the ReAct-style loop (Yao et al., 2023) for acting, but decouples the interaction history from epistemic reasoning.

We evaluate SCOUT on the challenging benchmarks, including ∞BENCH and LOOGLE-V2. Experimental results demonstrate that SCOUT achieves state-of-the-art accuracy on multi-hop aggregation tasks while reducing token consumption by up to $8\times$ compared to full-context models. More importantly, SCOUT maintains near-constant performance as context length scales to 2M tokens, effectively alleviating the LTU trilemma in practice.

---

[1]Efficiency here refers to *token cost*, not wall-clock latency; see Appendix A and Appendix F.4.

Our contributions are summarized as follows:

- We formalize the **LTU Trilemma**, identify task-agnostic processing as its primary bottleneck, and propose **SCOUT**, an active foraging paradigm that operates directly over raw documents without pre-indexing or chunking.

- We introduce a decoupled Epistemic State with provenance anchoring, which separates exploration traces from the reasoning substrate and makes long-horizon foraging reliable.

- We show that SCOUT matches or exceeds frontier long-text systems on complex reasoning while significantly reducing inference cost.

## 2. Methodology

In this section, we present SCOUT, a paradigm for million-token LTU that forages on demand for query-relevant information. We begin with the information sparsity assumption and a *Partially Observable Markov Decision Process* (POMDP) view, which motivate query-conditioned acquisition rather than exhaustive processing (Section 2.1). We then show that monolithic ReAct-style history-as-state agents (Yao et al., 2023) do not naturally satisfy three LTU requirements: relevance alignment, progress awareness, and sufficiency control (Section 2.2). Finally, we present SCOUT, which decouples the agent state into a procedural trace and a compact, provenance-anchored epistemic state, progressively refining the latter until it converges toward query sufficiency (Section 2.3).

### 2.1. Information Sparsity and the POMDP View

**Information sparsity as a practical reality.** Million-token LTU exhibits a mismatch between token volume and information density: for any query, only a small subset of the document is needed for answering. Let $\mathcal{D}$ be a document with length $N$ far exceeding the model's context budget (e.g., $N \geq 2M$), and let $\mathcal{F}(\mathcal{D})$ denote its atomic information units (e.g., facts or propositions).

For a query $q$, we posit an **oracle sufficient information set** $\mathcal{F}_q^\star \subseteq \mathcal{F}(\mathcal{D})$, defined as the smallest subset that preserves answerability compared to the full context (with $y$ the answer and $P(\cdot \mid q, \cdot)$ the answer distribution induced by the underlying model):

$$\mathcal{F}_q^\star = \underset{\mathcal{F}' \subseteq \mathcal{F}(\mathcal{D})}{\arg\min} \{|\mathcal{F}'| : P(y \mid q, \mathcal{F}') = P(y \mid q, \mathcal{F}(\mathcal{D}))\}. \tag{1}$$

The **Information Sparsity Assumption** states that, in long-context settings,

$$|\mathcal{F}_q^\star| \ll |\mathcal{F}(\mathcal{D})|. \tag{2}$$

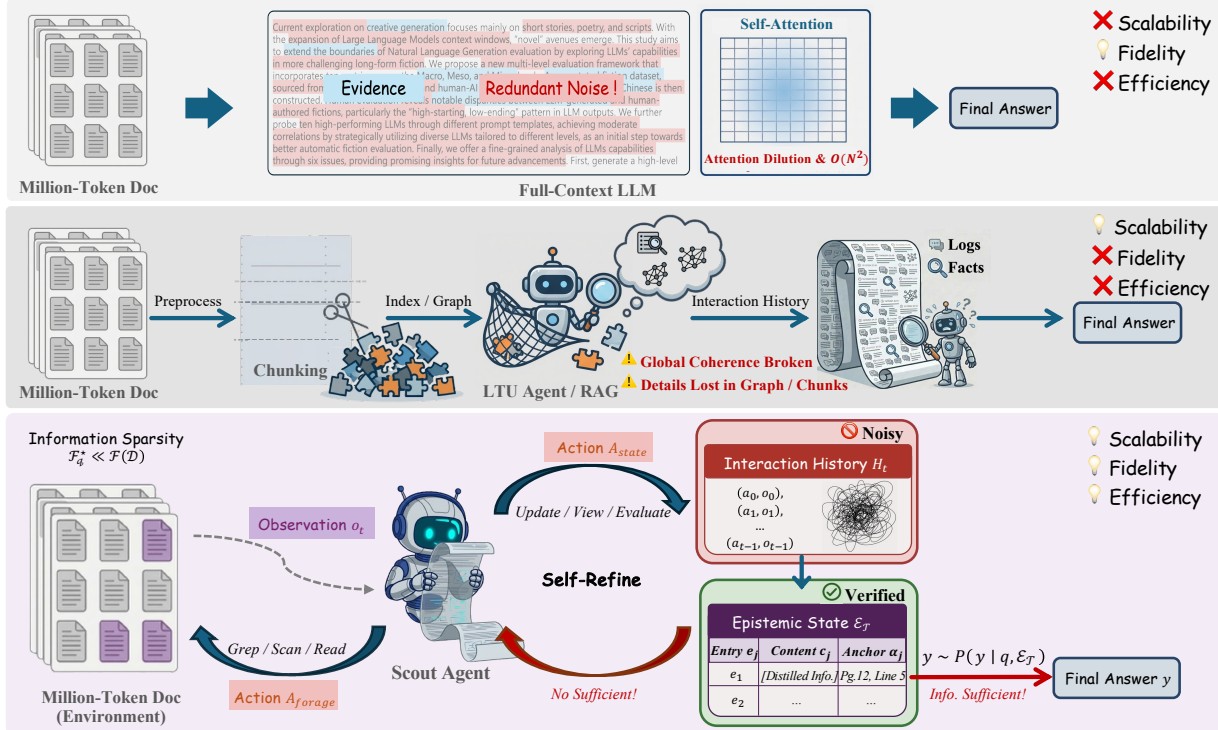

*Figure 2.* **Paradigm comparison for million-token LTU and SCOUT's state-decoupled convergence. Top:** Full-context LLMs ingest the entire sequence, suffering attention dilution and (dense-attention) quadratic compute/memory growth, so scalability and efficiency degrade under million-token inputs. **Middle:** Chunking/RAG-style agents improve scalability via fragmented access, but coherence can break across long ranges and answer-critical details may be lost in graphs/chunks, hurting fidelity. **Bottom (ours):** SCOUT treats the document as an explorable environment and decouples procedural exploration (a noisy interaction history) from epistemic reasoning (a distilled state). Across steps, the agent distills anchored knowledge into $\mathcal{E}_t$ and uses state-level diagnosis to guide foraging, progressively contracting $\mathcal{E}_t$ toward $\mathcal{F}_q^\star$.

**From exhaustive ingestion to key-information acquisition.** Equation (2) implies a simple design principle: the agent may need broad access to locate query-relevant information, but the amount of information retained for reasoning should remain sparse. Rather than ingesting the entire document, our objective is to acquire and consolidate a minimal, query-sufficient subset $\mathcal{F}_q^\star$.

**Modeling LTU Tasks as a POMDP.** We view LTU as sequential information acquisition under partial observability. Because the full document $\mathcal{D}$ cannot be processed in one pass, the agent only observes local fragments per step; consequently, decisions rely on an evolving interaction history, motivating a POMDP formulation.

Formally, a Partially Observable Markov Decision Process (POMDP) is defined as $\mathcal{M} = \langle \mathcal{S}, \mathcal{A}, \Omega, \mathcal{T}, \mathcal{R} \rangle$, where $\mathcal{S}$ denotes the latent state space, $\mathcal{A}$ the action space, $\Omega$ the observation space, $\mathcal{T}$ the transition dynamics, and $\mathcal{R}$ the reward function. In our LTU setting, $s \in \mathcal{S}$ denotes a complete task instance (the full document $\mathcal{D}$ and query $q$). At step $t$, an action $a_t \in \mathcal{A}$ is taken and a local observation $o_t \in \Omega$ is returned. While $\mathcal{D}$ is static, $\mathcal{T}$ governs how the agent–environment interaction evolves under actions. Since $s$ is latent, decisions rely on a belief state: a posterior over $s$ updated from the interaction history and incoming observations. Formal definitions are provided in Appendix C.1.

### 2.2. History-as-State and Long-Horizon Requirements

A common operationalization of on-demand key-information acquisition (Section 2.1) is an interactive agent that sequentially selects information-access actions over a long document $\mathcal{D}$, obtaining local observations and using the accumulated interaction to guide future decisions. This is a direct instantiation of the POMDP view under partial observability, and is often implemented in the ReAct-style (Yao et al., 2023). Let the interaction log be $\mathcal{H}_t = \{(a_0, o_0), \ldots, (a_{t-1}, o_{t-1})\}$, where $a_t$ denotes an action and $o_t$ is the returned observation. A standard **history-as-state** instantiation uses the growing history $\mathcal{H}_t$ as the sole state representation for both acting and answering, where $T$ denotes the terminal step:

$$a_t \sim \pi(a \mid q, \mathcal{H}_t), \qquad y \sim P(y \mid q, \mathcal{H}_T). \quad (3)$$

In other words, the same context serves as (1) a procedural trace that supports exploration control and (2) the reasoning input used to synthesize the final response.

**Structural requirements under information sparsity.** A monolithic history-as-state trajectory $\mathcal{H}_t$ is often adequate for short-horizon tool use, where the agent can largely collect more and answer from a compact interaction log. In LTU, however, information sparsity (Equation (2)) shifts the optimization target from *collecting* to *isolating the oracle information set* $\mathcal{F}_q^\star$. As a result, most observations acquired during exploration fall outside $\mathcal{F}_q^\star$ and act as noise for answering $q$. This makes a vanilla history-as-state instantiation brittle and inefficient: as $\mathcal{H}_t$ grows, the signal-to-noise ratio decreases while the model must still condition on the full history. Compression-based variants (e.g., ReSum (Wu et al., 2025), DeepAgent (Li et al., 2025c)) shorten the context via lossy compression, but may hurt fidelity by discarding position-anchored cues.

Consequently, an LTU agent should satisfy three requirements: **(R1) Relevance alignment:** continuously filter noisy observations to extract information consistent with $\mathcal{F}_q^\star$, and preserve it in a stable form for downstream reasoning; **(R2) Progress awareness:** track what has been established and what remains missing to guide subsequent exploration; **(R3) Sufficiency control:** assess query sufficiency and decide whether to continue or terminate.

## 2.3. The SCOUT Paradigm

To meet these requirements in Section 2.2, SCOUT reframes LTU as goal-directed information foraging under partial observability rather than exhaustive full-context processing.

We present SCOUT via three components. Section 2.3.1 introduces the core shift, **state decoupling**, which separates exploration from reasoning and enforces *state-only* answering to resist exploration noise (R1). Section 2.3.2 specifies Support I: provenance-anchored epistemic units for verifiable, high-fidelity state updates (R1). Section 2.3.3 presents Support II: gap-diagnosed epistemic convergence for progress tracking and sufficiency control (R2–R3).

### 2.3.1. CORE: STATE DECOUPLING AND EPISTEMIC STATE

Formally, SCOUT introduces state decoupling by separating the ReAct-style interaction history into two roles: a procedural trace $\mathcal{H}_t$ records *where the agent explores*, and a separate **epistemic state** $\mathcal{E}_t$ for reasoning that records *what has been established* as query-relevant knowledge.

Accordingly, the policy may condition on both to retain procedural awareness (e.g., avoiding redundant actions):

$$\pi(a_t \mid q, \mathcal{H}_t, \mathcal{E}_t).$$

---

**Algorithm 1** SCOUT Inference (state evolution with $\mathcal{E}_t$)

---
**Require:** Document $\mathcal{D}$, Query $q$, Max steps $T_{\max}$, Policy $\pi_\theta$
**Ensure:** Answer $y$
1: **Initialize:** $\mathcal{H}_0 \leftarrow \emptyset$, $\mathcal{E}_0 \leftarrow \emptyset$, $g_0 \leftarrow \perp$, $t \leftarrow 0$
2: **while** $t < T_{\max}$ **and** $g_t \neq \emptyset$ **do**
3:     Sample action $a_t \sim \pi_\theta(a \mid q, \mathcal{H}_t, \mathcal{E}_t)$
4:     **if** $a_t \in \mathcal{A}_{\text{forage}}$ **then**
5:         $o_t \leftarrow \text{ENV}(\mathcal{D}, a_t)$     ◁ *acquire local observation*
6:         $\mathcal{E}_{t+1} \leftarrow \mathcal{E}_t$,  $g_{t+1} \leftarrow g_t$     ◁ *state unchanged*
7:     **else**
8:                                         ◁ $a_t \in \mathcal{A}_{\text{state}}$
9:         **if** $a_t = \text{Update}$ **then**
10:           $\mathcal{E}_{t+1} \leftarrow \text{COMMIT}(\mathcal{E}_t, \mathcal{H}_t)$
11:           $o_t \leftarrow \mathcal{E}_{t+1}$   ◁ *commit distilled/anchored epistemic unit(s)*
12:           $g_{t+1} \leftarrow g_t$
13:         **else if** $a_t = \text{Evaluate}$ **then**
14:           $g_{t+1} \leftarrow \text{DIAGNOSEGAP}(q, \mathcal{E}_t)$  ◁ *update epistemic gap w.r.t.* $\mathcal{F}_q^\star$, $g_t$: *epistemic gap*
15:           $o_t \leftarrow g_{t+1}$
16:           $\mathcal{E}_{t+1} \leftarrow \mathcal{E}_t$
17:         **else**
18:           $o_t \leftarrow \text{APPLYSTATEOP}(\mathcal{E}_t, \mathcal{H}_t, a_t)$   ◁ *other actions*
19:           $\mathcal{E}_{t+1} \leftarrow \mathcal{E}_t$,  $g_{t+1} \leftarrow g_t$
20:         **end if**
21:     **end if**
22:     $\mathcal{H}_{t+1} \leftarrow \mathcal{H}_t \cup \{(a_t, o_t)\}$     ◁ *trace always grows*
23:     $t \leftarrow t + 1$
24: **end while**
25: **Decoupled reasoning:** $y \sim P(y \mid q, \mathcal{E}_t)$  ◁ *no access to* $\mathcal{H}_t$
26: **return** $y$

---

Crucially, SCOUT enforces a hard information-flow constraint at the answer step:

$$P(y \mid q, \mathcal{H}_T) \xrightarrow{\text{SCOUT}} P(y \mid q, \mathcal{E}_T), \qquad (4)$$

i.e., the final response is produced without access to the full interaction trace $\mathcal{H}_T$. This directly targets (R1): it prevents sparsity-induced exploration noise in $\mathcal{H}_T$ from contaminating reasoning, and makes the cognitive load scale with $|\mathcal{E}_T|$ rather than the raw document length $|\mathcal{D}|$.

### 2.3.2. SUPPORT I: PROVENANCE-ANCHORED EPISTEMIC UNITS

Since SCOUT performs reasoning over a decoupled epistemic state, $\mathcal{E}_t$ cannot be an unconstrained free-form state. We represent $\mathcal{E}_t$ as a set of provenance-anchored units, $\mathcal{E}_t = \{e_1, \ldots, e_{M_t}\}$, where each entry is $e_j = \langle c_j, \alpha_j \rangle$. Here $c_j$ is a distilled atomic statement and $\alpha_j$ is a provenance anchor (e.g., span identifiers) that points to a unique region in $\mathcal{D}$.

Operationally, the agent updates $\mathcal{E}_t$ only when it commits a unit derived from recent observations; purely exploratory actions leave $\mathcal{E}_t$ unchanged. Anchoring makes $\mathcal{E}_t$ auditable: any statement used in reasoning can be traced back to the source text, discouraging ungrounded abstractions.

### 2.3.3. SUPPORT II: GAP-DIAGNOSED EPISTEMIC CONVERGENCE

Under the Information Sparsity Assumption (Equation (2)), the objective of exploration is not to maximize coverage of $\mathcal{D}$, but to drive the epistemic state toward the query-sufficient content. SCOUT therefore views interaction as an *epistemic convergence* process: through iterative updates, $\mathcal{E}_t$ is pushed to approach the (unknown) oracle sufficient set $\mathcal{F}_q^\star$.

**State-level diagnosis for convergence.** At each step, the agent alternates between acquiring new observations and committing anchored epistemic units into $\mathcal{E}_t$. Crucially, SCOUT applies a state-level assessment operator (Evaluate) to the current $\mathcal{E}_t$ to diagnose its gap to $\mathcal{F}_q^\star$. Rather than serving as a termination heuristic, this diagnosis characterizes what is still missing for answering $q$ reliably, thereby providing an explicit target for the next round of foraging and commits. Iterating this diagnostic–acquire–commit loop yields a self-refinement dynamics that drives $\mathcal{E}_t$ to converge toward $\mathcal{F}_q^\star$:

$$\underbrace{\overset{\text{Diagnose \& Update}}{\circlearrowleft} \mathcal{E}_t}_{\text{Epistemic State}} \overset{\lim_{t \to T}}{\dashrightarrow} \underbrace{\mathcal{F}_q^\star}_{\text{Oracle Sufficient Set}} \tag{5}$$

In the idealized limit, this dynamics aims to make the committed state $\mathcal{E}_T$ contain a query-sufficient subset, i.e., $\mathcal{E}_T \approx \mathcal{F}_q^\star$ in the sense of preserving $P(y \mid q, \cdot)$ as in Equation (1). This gap diagnosis makes progress explicit, it identifies what is still missing in $\mathcal{E}_t$ and thereby guides subsequent foraging (R2). When the diagnosed gap is resolved, query sufficiency becomes a state property, enabling a principled completion criterion independent of the noisy trace $\mathcal{H}_t$ (R3).

### 2.3.4. IMPLEMENTATION: ACTION SPACE FOR FORAGING AND STATE MANAGEMENT

We instantiate the above loop with a composite action space $\mathcal{A} = \mathcal{A}_{\text{forage}} \cup \mathcal{A}_{\text{state}}$, decoupling physical document foraging from cognitive state maintenance.

- **Foraging actions ($\mathcal{A}_{\text{forage}}$):** multi-resolution exploration primitives over $\mathcal{D}$, ranging from lexical skimming (Grep, Scan) to dense local reading (Read).

- **Epistemic management actions ($\mathcal{A}_{\text{state}}$):** state operations over $\mathcal{E}_t$ that implement the decoupled epistemic workflow in Section 2.3. Specifically, Update commits new provenance-anchored units $\langle c, \alpha \rangle$ into $\mathcal{E}_t$ (Support I), Evaluate performs state-level sufficiency

assessment by diagnosing the remaining information gap of $\mathcal{E}_t$ with respect to $\mathcal{F}_q^\star$ (Support II).

More concretely, these actions are selected autonomously by the agent policy, rather than being enforced by a fixed pipeline. In practice, Evaluate is implemented as a **state-level tool** call that re-invokes the same backbone LLM with a constrained prompt and a fixed output schema to produce a machine-parsable gap diagnosis $g_t$ from $(q, \mathcal{E}_t)$

Algorithm 1 summarizes the inference loop. Action details are provided in Appendix C.2, and a trajectory example is presented in Appendix G.

## 3. Experiments

### 3.1. Setup

**Benchmarks.** We focus on benchmarks that require information synthesis beyond passive retrieval, and thus move beyond Needle-in-a-Haystack (NIAH) style tests. Specifically, we evaluate on two complementary benchmarks as follows; more details are provided in Appendix D.

- **LOOGLE-V2** (He et al., 2025): targets **long-horizon dependency reasoning**, requiring multi-hop synthesis across widely separated evidence rather than single-point lookup.

- **∞BENCH** (Zhang et al., 2024): targets **robustness at ultra-long scale** (up to 1M+ tokens) on real-world documents (e.g., novels, codebases), stress-testing whether models can locate sparse signals amid realistic distractors.

**Baselines.** In addition to frontier long-context models such as Gemini-3-Pro and GPT-5.1, which represent strong monolithic-context baselines, we compare SCOUT against specialized long-context agents reported in recent benchmarks. These include **ReadAgent** (Lee et al., 2024), which uses page-level gist memory with interactive look-up to handle very long documents; **GraphReader** (Li et al., 2024b), a graph-based navigation agent that performs coarse-to-fine exploration and shows strong performance on single-hop and multi-hop QA; **MemAgent** (Yu et al., 2025a), a memory-augmented framework that reads text in segments and updates a persistent memory state; and **Claude Code** (Anthropic, 2025b), a coding-oriented agent with file-system interaction that performs strongly on repository-level tasks. To isolate architectural effects from model strength, we adopt a **Unified Backbone Protocol**: all agentic baselines (ReadAgent, GraphReader, MemAgent, and SCOUT) use the same backbone (Claude-Sonnet-4.5). Alternative backbones are reported in Appendix F.1.

*Table 1.* **Main results on LOOGLE-V2 and ∞BENCH.** We compare SCOUT with frontier native long-context LLMs and recent agentic baselines. We report **accuracy** (%) and **token cost** (k tokens), and compute **Token Eff.** as Acc/Cost. SCOUT achieves the highest overall accuracy on both benchmarks while delivering the highest Token Eff., indicating a superior accuracy–cost tradeoff.

*(a)* LOOGLE-V2

| Category | Method | Accuracy (↑, %) | | | | | Efficiency | |
|---|---|---|---|---|---|---|---|---|
| | | Overall | Code | Finance | Game | Law | Cost (k) ↓ | Token Eff. ↑ |
| **LLMs** | GPT-4.1 (OpenAI, 2025a) | 56.8 | 48.1 | 61.0 | 46.3 | 72.9 | 236.0 | 0.24 |
| | Gemini-2.5-Pro (Comanici et al., 2025) | 56.2 | 25.6 | 56.2 | 68.6 | 79.1 | 258.1 | 0.22 |
| | Claude-Sonnet-4.5 (Anthropic, 2025a) | 42.5 | 26.0 | 45.7 | 38.8 | 62.0 | 100.9 | 0.42 |
| | GPT-5.1-chat (OpenAI, 2025b) | 58.6 | 40.9 | 66.7 | 51.7 | 76.7 | 135.9 | 0.43 |
| | Gemini-3-Pro (Google DeepMind, 2025) | 68.5 | 60.0 | 79.4 | 48.1 | **87.5** | 273.9 | 0.25 |
| **Agents** | ReadAgent (Lee et al., 2024) | 27.7 | 26.0 | 35.0 | 27.3 | 22.0 | 302.6 | 0.09 |
| | GraphReader (Li et al., 2024b) | 31.6 | 24.9 | 35.0 | 32.2 | 35.3 | 408.1 | 0.08 |
| | MemAgent (Yu et al., 2025a) | 46.0 | 35.7 | 49.8 | 46.4 | 53.3 | 302.2 | 0.15 |
| | Claude Code (Anthropic, 2025b) | 63.4 | 41.2 | 77.0 | 76.2 | 61.2 | **25.3** | 2.51 |
| **Ours** | SCOUT | **78.7** | **61.7** | **80.6** | **88.9** | 86.7 | 29.7 | **2.63** |

*(b)* ∞BENCH

| Category | Method | Accuracy (↑, %) | | | | | Efficiency | |
|---|---|---|---|---|---|---|---|---|
| | | Overall | Retrieval | Code | Math | Text | Cost (k) ↓ | Token Eff. ↑ |
| **LLMs** | GPT-4.1 (OpenAI, 2025a) | 73.0 | 99.2 | 47.3 | 60.6 | 43.6 | 212.7 | 0.34 |
| | Gemini-2.5-Pro (Comanici et al., 2025) | 81.9 | 96.2 | 63.4 | **97.9** | 58.5 | 245.4 | 0.33 |
| | Claude-Sonnet-4.5 (Anthropic, 2025a) | 65.1 | 97.6 | 33.3 | 12.9 | 46.5 | 96.6 | 0.67 |
| | GPT-5.1-chat (OpenAI, 2025b) | 82.6 | 92.0 | **88.2** | 92.3 | 51.5 | 118.2 | 0.70 |
| | Gemini-3-Pro (Google DeepMind, 2025) | 83.9 | 97.0 | 67.1 | 80.6 | **71.1** | 259.1 | 0.32 |
| **Agents** | ReadAgent (Lee et al., 2024) | 42.3 | 54.7 | 20.4 | 29.9 | 40.9 | 481.8 | 0.09 |
| | GraphReader (Li et al., 2024b) | 43.1 | 55.4 | 12.9 | 94.4 | 16.9 | 327.4 | 0.13 |
| | MemAgent (Yu et al., 2025a) | 68.1 | **99.5** | 24.2 | 76.6 | 33.0 | 264.0 | 0.26 |
| | Claude Code (Anthropic, 2025b) | 72.7 | 90.2 | 71.0 | 65.6 | 38.6 | **19.2** | 3.79 |
| **Ours** | SCOUT | **85.6** | 99.4 | 75.3 | 85.7 | 63.9 | 21.4 | **4.01** |

**Metrics.** We report accuracy (%) and total token cost (prompt/context plus generated tokens, summed across all inference turns). For agentic methods, this includes every turn of the multi-step interaction; for monolithic LLMs, it is the single-pass cost. We also compute Token Eff. as Acc/Cost to summarize the accuracy–cost tradeoff (higher is better). Full cost accounting is detailed in Appendix E.1. We additionally report wall-clock latency in Appendix F.4.

### 3.2. Main Results

Table 1 and Figure 1 compare native long-context LLMs and agentic frameworks on both LOOGLE-V2 and ∞BENCH. Extended results with open-weight LLMs (Qwen2.5, Qwen3) are provided in Appendix F.2.

**Takeaway:** Under the Unified Backbone Protocol, SCOUT achieves the highest overall accuracy on both LOOGLE-V2 and ∞BENCH and the best accuracy–cost tradeoff, substan-

tially outperforming prior agentic baselines while matching or exceeding frontier native long-context LLMs on LOOGLE-V2 with over an $8\times$ reduction in token cost compared to Gemini-3-Pro (achieving the highest Token Eff.).

### 3.3. Analysis: Alleviating the LTU Trilemma in Practice

As introduced in Section 1, LTU systems face a practical trilemma among **scalability** (robustness as context length grows), **information fidelity** (faithful reasoning, proxied by LOOGLE-V2 accuracy), and **inference efficiency** (token cost). Figure 3 visualizes this tradeoff by sweeping the available context length from 64K to 1M+ tokens and jointly reporting accuracy (left) and token cost (right).

**Takeaway:** SCOUT alleviates the trilemma by keeping fidelity and efficiency nearly flat as context grows. In Figure 3 (left), frontier monolithic long-context LLMs exhibit a clear long-horizon breakdown: accuracy peaks around shorter

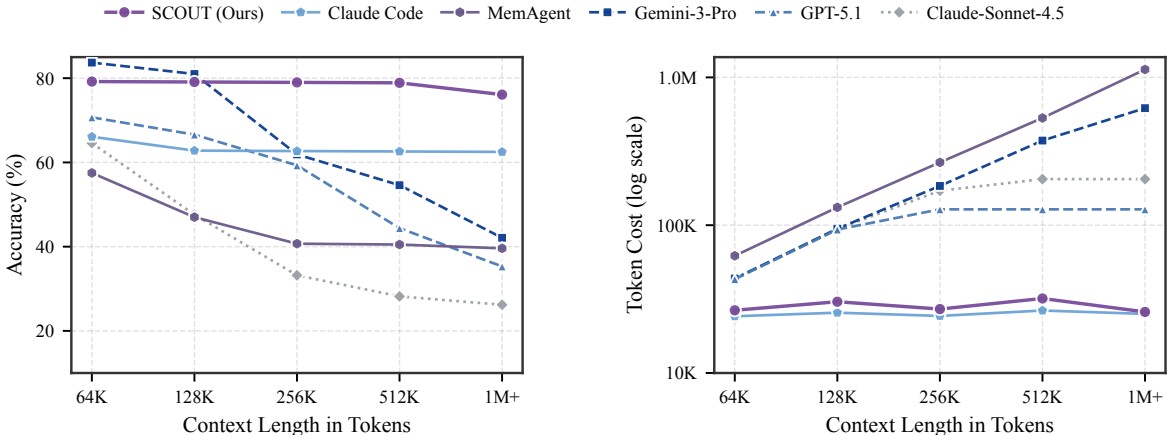

*Figure 3.* **Alleviating the LTU trilemma under scaling. Left**: as context grows from 64K to 1M+, SCOUT maintains near-flat, top accuracy, while frontier monolithic LLMs often degrade at long horizons. **Right** (log scale): SCOUT keeps token cost nearly constant across the sweep, whereas monolithic ingestion costs grow rapidly with context length. Together, SCOUT remains near the Pareto frontier, simultaneously achieving strong fidelity, efficiency, and scalability.

windows but degrades sharply as the context extends to 256K–1M+, indicating failures in maintaining long-range dependencies rather than simple window limitations. Meanwhile, Figure 3 (right) shows that their token cost increases by orders of magnitude with context length.

### 3.4. Ablation Study

We ablate key design choices in SCOUT to isolate the contributions of (i) the Epistemic State $\mathcal{E}_t$ with explicit grounding, and (ii) the hierarchical action space (Section 2.3.3) under a unified evaluation protocol. Table 2 reports LOOGLE-V2 results; ∞BENCH counterparts are provided in Appendix F.5 for completeness.

**Ablation variants.** w/o $\mathcal{E}_t$ (ReAct-style): remove the decoupled epistemic state and use the monolithic history $\mathcal{H}_t$ for both acting and answering throughout. w/o Foraging Actions ($\mathcal{A}_{\text{forage}}$): remove the information-access action set $\mathcal{A}_{\text{forage}}$ (cannot access $\mathcal{D}$; effectively forced to guess under severe information constraints). w/o File Tools: remove file-operation actions other than $\mathcal{A}_{\text{forage}}$, retaining only basic navigation primitives. w/o Grounding ($\alpha$): disable grounding in state update, updating epistemic units without explicit anchoring constraints or provenance references.

### 3.5. Open-Weight Models: Training Gains under SCOUT

We report open-weight results under controlled post-training budgets. In the **LLM setting**, SFT trains single-pass answering from a monolithic prompt (including the long context), while in the **SCOUT setting**, SFT/DAPO (Yu et al., 2025b) trains the backbone LLM as an action policy to decide tool actions and state updates, and to produce the

*Table 2.* **Paradigm ablation on LOOGLE-V2.** Removing $\mathcal{E}_t$ (ReAct-style) degrades long-horizon retention; removing $\mathcal{A}_{\text{forage}}$ collapses performance by preventing access to $\mathcal{D}$. Grounding and auxiliary file tools provide additional gains.

| Variant | Acc (%) | $\Delta$ | Cost (k) |
|---|---|---|---|
| **SCOUT (Full)** | **78.2** | – | **29.7** |
| w/o $\mathcal{E}_t$ (ReAct-style) | 70.5 | -7.7 | 28.3 |
| w/o $\mathcal{A}_{\text{forage}}$ (no access to $\mathcal{D}$) | 17.7 | -60.5 | 27.4 |
| w/o File Tools | 74.2 | -4.0 | 31.4 |
| w/o Grounding ($\alpha$) | 75.5 | -2.7 | 29.7 |

final answer. Across both the settings, the post-training throughput is matched to approximately 60M optimization tokens (prompt/context plus generated tokens used for learning); for DAPO, this includes rollout tokens consumed for policy updates. Full training details (data construction, budget-matching protocol, and RL setup) are provided in Appendix E.3.

**Takeaway: The same budget pays off only when paired with the SCOUT interface.** Two observations stand out from Table 3. *(i) LLM-only SFT is not the lever for million-token LTU:* when the backbone is trained and evaluated *as a monolithic LLM*, SFT yields only marginal gains and can even be neutral under a fixed budget. *(ii)* SCOUT *turns lightweight post-training into large LTU gains:* training and evaluating the same backbone within the SCOUT paradigm produces substantially larger improvements under the same optimization-token budget, and DAPO further amplifies these gains under the identical interaction protocol. Together, this gap indicates that the dominant bottleneck for open-weight backbones in million-token LTU is not raw parametric knowledge, but long-horizon interaction and

*Table 3.* **Open-weight post-training gains under SCOUT.** With controlled post-training, LLM-only SFT brings little to no improvement, while SCOUT-based SFT/DAPO yields large gains on benchmarks. Each cell reports Acc (%) with (Δ vs. Vanilla)

| Backbone | Regime | LOOGLE-v2 | | ∞BENCH | |
|---|---|---|---|---|---|
| | | LLM | SCOUT | LLM | SCOUT |
| Qwen2.5-72B-Instruct (Qwen Team, 2024) | Vanilla | 24.0 | 28.3 | 49.4 | 53.2 |
| | +SFT | 25.1 (+1.2) | **55.4** (+27.1) | 48.7 (−0.7) | **78.3** (+25.1) |
| | +DAPO | – | 35.1 (+6.8) | – | 60.8 (+7.6) |
| Qwen3-30B-A3B-Instruct (Yang et al., 2025) | Vanilla | 40.8 | 41.2 | 55.5 | 65.4 |
| | +SFT | 40.3 (−0.5) | **54.2** (+13.0) | 57.1 (+1.6) | **75.8** (+10.4) |
| | +DAPO | – | 45.1 (+3.9) | – | 68.8 (+3.4) |

state control; SCOUT supplies the missing structure, allowing comparatively weak open models to approach the performance regime of strong closed-source systems on LTU. More analysis is provided in Appendix F.6.

## 4. Related Work

We situate our work in Long-Text Understanding (LTU), covering monolithic long-context models, specialized agents, and structured retrieval.

### 4.1. Large-Scale Context Modeling

Ingesting massive contexts has defined the 2026 LLM landscape. Leading proprietary models, including Gemini-3-Pro (Google DeepMind, 2025), have standardized the 1 million token window. Similarly, GPT-5 (OpenAI, 2025b) and Claude-Sonnet-4.5 (Anthropic, 2025a) now support ultra-long contexts of 512k tokens. In the open-weight domain, Qwen3-235B-A22B-Thinking (Yang et al., 2025) has successfully scaled to 256k tokens while incorporating reasoning-specific optimizations. These models demonstrate near-perfect fidelity on standard Needle-in-a-Haystack (NIAH) retrieval tasks, creating an illusion that the long-context problem is solved. However, the research community has increasingly shifted focus from simple retrieval to Long-Range Dependency and Information Aggregation. ∞BENCH (Zhang et al., 2024) shows that accuracy drops sharply beyond 100k tokens when key information is implicit. LOOGLE-v2 (He et al., 2025) further exposes failures in Multi-hop Aggregation, where answers require synthesizing evidence dispersed across the document.

### 4.2. Specialized Long-Context Agents

To bypass context limits without retraining, recent research explores agentic decomposition, which generally falls into two paradigms: linear decomposition and structured navigation. **Linear and Collaborative Decomposition.** Approaches like LongAgent (Zhao et al., 2024) and Chain of Agents (Li et al., 2025b) scale context handling via

hierarchical or relay-based hand-offs. Memory-augmented methods like MemAgent (Yu et al., 2025a), Self-Note (Lanchantin et al., 2023), and generative-associative memory (Zhang et al., 2026; Jia et al., 2025) utilize external scratchpads to record states across segments. While effective for parallel extraction, these methods function primarily as passive "readers," constrained by rigid linear workflows and prone to information forgetting during inter-agent transitions. **Structured Navigation.** Moving beyond linear reading, frameworks such as GraphReader (Li et al., 2024b) and ReadAgent (Lee et al., 2024) convert documents into entity graphs or gist indices for non-linear navigation, but require heavy pre-computation. Graph abstractions also create a bottleneck of abstraction: details missed during structure construction become irretrievable. Even ReadAgent retains original pages but still pays substantial upfront indexing overhead before foraging.

### 4.3. Structured Retrieval and Agentic RAG

Retrieval-Augmented Generation (RAG) mitigates context costs by retrieving text segments via vector similarity. Recent Agentic RAG paradigms (Singh et al., 2025) evolve this by incorporating active planning, reflection (Asai et al., 2024), structured indices like trees and graphs (Guo et al., 2025; Sarthi et al., 2024; Qian et al., 2024), or adaptive memory that learns from user feedback (Li et al., 2024a) to refine retrieval quality. Nevertheless, these approaches remain limited by an index-first design: they build task-agnostic abstractions before the query is known. As Edge et al. (2024) argues, this fragmentation disrupts global coherence and long-range dependencies, making such methods ill-suited for reasoning that relies on document-wide interactions.

## 5. Conclusion

We revisited long-text understanding through the LTU trilemma and argued for a shift from brute-force ingestion (or task-agnostic, pre-abstracted chunking) to an *active information foraging* paradigm: treat the document as an explorable environment, while keeping the reasoning substrate

sparse and query-sufficient. SCOUT realizes this paradigm through decoupled epistemic states and convergence-driven foraging. Empirically, it achieves a better accuracy–cost trade-off and stable performance at million-token scale, mitigating the LTU trilemma in practice.

## Impact Statement

This work aims to advance machine learning for long-text understanding at extreme context lengths by improving the accuracy–efficiency trade-off. The primary benefits are reduced computational cost (and potentially energy use) for long-document analysis in scientific and technical applications. Potential risks are mainly from misuse on sensitive or proprietary texts and from over-reliance on imperfect model outputs in high-stakes settings. We recommend standard mitigations, including access control and data governance for private documents, and human verification for consequential decisions.

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

*Table 4.* **Summary of notation used in Section 2.**

| Symbol | Type | Meaning |
|---|---|---|
| *Problem Setup (Section 2.1)* | | |
| $\mathcal{D}, q, y$ | – | Document, query, answer |
| $\mathcal{F}(\mathcal{D})$ | set | Set of atomic information units extractable from $\mathcal{D}$ |
| $\mathcal{F}_q^\star$ | set | Oracle sufficient information set; minimal subset preserving answerability |
| $\mathcal{M}$ | tuple | POMDP $\langle \mathcal{S}, \mathcal{A}, \Omega, \mathcal{T}, \mathcal{R} \rangle$ (see Appendix C.1) |
| $\mathcal{S}, \Omega$ | set | Latent state space; observation space |
| $\mathcal{T}, \mathcal{R}$ | function | Transition dynamics; reward function |
| *Interaction Dynamics (Sections 2.2 and 2.3)* | | |
| $a_t$ | action | Action selected at step $t$ |
| $o_t$ | observation | Observation returned at step $t$ (local view of $\mathcal{D}$) |
| $\mathcal{H}_t$ | history | Interaction history $\{(a_0, o_0), \dots, (a_{t-1}, o_{t-1})\}$ |
| $T_{\max}$ | scalar | Maximum interaction steps (budget) |
| SCOUT *Paradigm (Section 2.3)* | | |
| $\mathcal{E}_t$ | state | Epistemic State at step $t$; set of verified evidence entries |
| $e_j = \langle c_j, \alpha_j \rangle$ | entry | Single epistemic entry: content $c_j$ + provenance anchor $\alpha_j$ |
| $c_j$ | text | Distilled content of entry $e_j$ |
| $\alpha_j$ | anchor | Provenance anchor linking $c_j$ to source location in $\mathcal{D}$ |
| $g_t$ | diagnosis | Gap diagnosis at step $t$; identifies missing information w.r.t. $\mathcal{F}_q^\star$ |
| $\mathcal{A}$ | set | Action space = $\mathcal{A}_{\text{forage}} \cup \mathcal{A}_{\text{state}}$ |
| $\mathcal{A}_{\text{forage}}$ | set | Foraging actions: `Grep`, `Scan`, `Read`, etc. |
| $\mathcal{A}_{\text{state}}$ | set | Epistemic management: `Update`, `View`, `Evaluate` |
| $\pi_\theta$ | policy | Foraging policy; $\pi_\theta(a_t \mid q, \mathcal{H}_t, \mathcal{E}_t, g_t)$ |
| $\tau$ | trajectory | Full interaction trajectory induced by $\pi_\theta$ |

# A. Limitations

While SCOUT achieves strong accuracy–cost trade-offs, several limitations should be noted. First, SCOUT relies on multi-step interaction with repeated LLM calls, which results in higher wall-clock latency than single-pass monolithic models (see Appendix F.4); although its latency remains near-constant as context length scales, the absolute per-query latency is higher, and latency optimization (e.g., parallel foraging, speculative actions) remains an important direction for future work. Second, SCOUT currently processes each query independently: when multiple queries target the same document, the agent re-explores from scratch without reusing previously acquired knowledge, whereas index-based or RAG methods amortize their pre-processing cost across queries; extending the epistemic state to support cross-query reuse is a promising direction. Third, our evaluation focuses on text-only, information-sparse long documents where query-relevant evidence constitutes a small fraction of the total content, which aligns well with SCOUT's design assumptions; the applicability to information-dense settings (where most of the document is relevant) and to multimodal long documents (e.g., interleaved text-image-table documents) remains to be investigated.

# B. Notation

This table summarizes the core symbols introduced in the methodology. $\mathcal{F}_q^\star$ is a theoretical construct (the minimal evidence needed to answer $q$); in practice, the agent aims to make $\mathcal{E}_T$ approximate $\mathcal{F}_q^\star$ through iterative foraging and gap diagnosis. The provenance anchor $\alpha_j$ ensures every piece of retained evidence can be traced back to its original location in $\mathcal{D}$, supporting auditability and reducing hallucination risk.

# C. Methodology Details

## C.1. Formal Problem Formulation (POMDP)

This section provides the detailed mathematical formulation of the Information Foraging POMDP introduced in Section 2.1, and clarifies why we adopt a POMDP rather than an MDP formulation.

**Why POMDP instead of MDP?**   In a standard Markov Decision Process (MDP), the agent has full access to the current state $s_t$ at every time step, and can therefore make decisions based on complete information. However, million-token LTU fundamentally violates this assumption: the "state" (the full document $\mathcal{D}$ plus query $q$) far exceeds the model's context window, so the agent can never observe $s$ in its entirety. Instead, each action yields only a *partial observation* $o_t$ (e.g., a few pages or search results), leaving most of the document unobserved.

This partial observability has critical implications for agent design:

1. **Decisions must be based on a belief state**, not the true state. The agent maintains a posterior belief over what information the document contains, updated as new observations arrive.

2. **Exploration is necessary**: since relevant evidence may be located anywhere in $\mathcal{D}$, the agent must actively search rather than passively receive information.

3. **History matters**: unlike MDPs where the current state is Markovian, in POMDPs the agent's decision quality depends on its accumulated interaction history $\mathcal{H}_t$.

**Belief State in LTU.**   In classical POMDP theory, the belief state $b_t$ is a probability distribution over latent states, updated via Bayes' rule as observations arrive. In our LTU setting, we use a *sufficient statistic* representation: the interaction history $\mathcal{H}_t = \{(a_0, o_0), \ldots, (a_{t-1}, o_{t-1})\}$ serves as a summary of all information the agent has gathered. SCOUT further refines this by decoupling the belief into two components:

- $\mathcal{H}_t$: the raw procedural trace (what actions were taken and what was observed), used for navigation and avoiding redundant exploration;

- $\mathcal{E}_t$: the **Epistemic State**, a compact, provenance-anchored representation of verified evidence, used for reasoning and answering.

This decoupling addresses the key challenge that $\mathcal{H}_t$ grows linearly with interaction length and becomes increasingly noisy, while $\mathcal{E}_t$ remains compact and query-focused.

**POMDP Tuple Definition.**   The process is defined by the tuple $\mathcal{M} = \langle \mathcal{S}, \mathcal{A}, \Omega, \mathcal{T}, \mathcal{R} \rangle$:

- **State Space ($\mathcal{S}$):** The latent global state $s \in \mathcal{S}$ corresponds to the fixed environment configuration: the full document corpus $\mathcal{D}$ and the user query $q$. In our setting, $s$ is static but fully unobservable to the agent due to context window constraints.

- **Action Space ($\mathcal{A}$):** Let $\mathcal{A}$ be a set of information acquisition primitives. At step $t$, an action $a_t \in \mathcal{A}$ functions as a query operator on the global state. This formulation generalizes various retrieval operations, from keyword search (Grep) to reading specific chunks (Read).

- **Observation Space ($\Omega$):** The observation $o_t \in \Omega$ is the partial information returned by the environment—a projection of the global state $s$ conditioned on action $a_t$. The core challenge of LTU arises from the information sparsity assumption (Equation (2)): the subset of $\Omega$ relevant to $q$ is extremely small relative to the size of $\mathcal{S}$.

- **Transition Dynamics ($\mathcal{T}$):** Unlike traditional RL where the environment state evolves stochastically ($s_t \rightarrow s_{t+1}$), the document state in LTU is invariant ($s_{t+1} = s_t = s$). However, the *agent's belief state* evolves deterministically as information accumulates: $\mathcal{H}_{t+1} = \mathcal{H}_t \cup \{(a_t, o_t)\}$.

- **Reward Function ($\mathcal{R}$):** The reward is sparse and terminal: the agent receives $R(\mathcal{H}_T) = \mathbb{I}(y = y^*)$ based on answer correctness. While token cost is an important practical consideration (reported in experiments), we do not incorporate an explicit step-wise penalty; instead, efficiency emerges from the agent's learned ability to terminate early once $\mathcal{E}_t$ is query-sufficient.

*Table 5.* **Action Space.**

| Action | Key parameters (short) | Returns (short) |
|---|---|---|
| *Foraging / environment actions ($\mathcal{A}_{forage}$)* | | |
| Glob | pattern (required): file pattern for locating candidate sources; scope (optional): where to search | Candidate file paths or source identifiers. |
| Grep | pattern (required): keyword/regex for lexical skimming; context (optional): small surrounding window size; scope (optional): where to search within $\mathcal{D}$ | Matched snippets with coarse provenance anchors (to support subsequent Read). |
| Read | anchor (required): provenance anchor (e.g., file + line range / offset); window (optional): read window size (e.g., offset/limit) | A contiguous local observation $o_t$ (dense context) grounded at the specified anchor. |
| Scan | pattern (required): lightweight structural pattern (e.g., headings/tables) to quickly locate candidate regions; scope (optional): where to scan within $\mathcal{D}$ | Candidate locations (anchors) to inspect next (e.g., matched headings/line ranges). |
| GetFileInfo | source (required): source identifier to inspect | Source metadata (e.g., size / estimated length) for budgeting. |
| *Epistemic management actions ($\mathcal{A}_{state}$) over $\mathcal{E}_t$* | | |
| Update | $(c, \alpha)$ (required): content $c$ and provenance anchor $\alpha$ to commit into $\mathcal{E}_t$ | Updated Epistemic State $\mathcal{E}_{t+1}$ (with the new entry committed). |
| View | $E\_id$ (required): which $\mathcal{E}_t$ to inspect | Current Epistemic State content (full $\mathcal{E}_t$). |
| Evaluate | $(q, \mathcal{E}_t)$ (required): query $q$ and current epistemic state $\mathcal{E}_t$ | Gap diagnosis $g_t$ (empty iff query-sufficient), optionally with a brief rationale. |
| *Auxiliary utilities (not part of $\mathcal{A}$ in Algorithm 2)* | | |
| CountTokens | text (required): text to tokenize; model (optional): tokenizer name | Token count for the given text. |
| TodoWrite | todos (required): list of items with content/status | Updated todo list. |
| NormalizeDocument | source (required): source identifier to normalize; max_length (optional): normalization granularity | Normalized source plus confirmation status. |

## C.2. Action Space Specification

Table 5 summarizes the action interface used by SCOUT. Consistent with the main text, the overall action space is decomposed into two categories: (i) **foraging / environment actions** $\mathcal{A}_{\text{forage}}$, which support lightweight navigation and local reading over the raw document (e.g., `Scan`/`Glob`/`Grep` for coarse localization and lexical skimming, and `Read` for anchored dense access, with optional budgeting utilities such as `GetFileInfo`); and (ii) **epistemic management actions** $\mathcal{A}_{\text{state}}$, which operate on the epistemic state $\mathcal{E}_t$ (e.g., `Update` to commit a grounded $(c, \alpha)$ unit, `View` to inspect $\mathcal{E}_t$, and `Evaluate` to diagnose the remaining epistemic gap $g_t$ and determine query sufficiency). In addition, we include a small set of **auxiliary utilities** (e.g., `CountTokens`, `TodoWrite` and `NormalizeDocument`) that facilitate implementation and bookkeeping but are not treated as part of the policy action space in Algorithm 2.

**Document representation in practice.** In our implementation, each long input is stored directly as a single `.txt` file, and the agent treats this file as the exploration target. We do *not* apply indexing or any document preprocessing: *no* **pre-indexing,** *no* **chunking, and** *no* **summarization**, so the framework can be applied broadly to raw long-text inputs while keeping the interaction semantics simple and transparent.

## C.3. SCOUT Inference Procedure

Algorithm 2 presents the complete inference procedure of the SCOUT paradigm. The key design principle is the **structural decoupling** between information foraging (exploration) and final reasoning: the answer generation depends exclusively on the distilled Epistemic State $\mathcal{E}_T$, strictly isolating the reasoning process from the noisy exploration history $\mathcal{H}_T$.

**Algorithm 2** SCOUT Inference Framework (Detailed)

---

**Require:** Document $\mathcal{D}$, Query $q$, Max steps $T_{\max}$, Policy $\pi_\theta$
**Ensure:** Answer $y$
1: **Initialize:** $\mathcal{H}_0 \leftarrow \emptyset$, $\mathcal{E}_0 \leftarrow \emptyset$, $g_0 \leftarrow \bot$, $t \leftarrow 0$
2: **while** $t < T_{\max}$ **and** $g_t \neq \emptyset$ **do**
3:      Sample action $a_t \sim \pi_\theta(a \mid q, \mathcal{H}_t, \mathcal{E}_t)$
4:      **if** $a_t \in \mathcal{A}_{\text{forage}}$ **then**
5:          $o_t \leftarrow \text{ENV}(\mathcal{D}, a_t)$ ◁ *execute foraging action (Glob/Grep/Read/Scan/GetFileInfo) on $\mathcal{D}$, return local observation with provenance anchors*
6:          $\mathcal{E}_{t+1} \leftarrow \mathcal{E}_t$,   $g_{t+1} \leftarrow g_t$              ◁ *foraging does not modify epistemic state; only acquires raw observations*
7:      **else**
8:                                         ◁ $a_t \in \mathcal{A}_{state}$: *epistemic management actions*
9:          **if** $a_t = \text{Update}$ **then**
10:              $\mathcal{E}_{t+1} \leftarrow \text{COMMIT}(\mathcal{E}_t, \mathcal{H}_t)$ ◁ *distill and commit anchored epistemic unit(s) $(c, \alpha)$ from recent observations into $\mathcal{E}_t$*
11:              $o_t \leftarrow \mathcal{E}_{t+1}$                        ◁ *return updated epistemic state for verification*
12:              $g_{t+1} \leftarrow g_t$
13:          **else if** $a_t = \text{Evaluate}$ **then**
14:              $g_{t+1} \leftarrow \text{DIAGNOSEGAP}(q, \mathcal{E}_t)$   ◁ *diagnose remaining epistemic gap w.r.t. query-sufficient set $\mathcal{F}_q^\star$; returns $\emptyset$ if sufficient*
15:              $o_t \leftarrow g_{t+1}$                      ◁ *gap diagnosis guides next foraging direction*
16:              $\mathcal{E}_{t+1} \leftarrow \mathcal{E}_t$
17:          **else if** $a_t = \text{View}$ **then**
18:              $o_t \leftarrow \text{RENDER}(\mathcal{E}_t)$     ◁ *inspect current epistemic state; useful for long-horizon planning and self-monitoring*
19:              $\mathcal{E}_{t+1} \leftarrow \mathcal{E}_t$,   $g_{t+1} \leftarrow g_t$
20:          **else**
21:              $o_t \leftarrow \text{APPLYSTATEOP}(\mathcal{E}_t, \mathcal{H}_t, a_t)$             ◁ *other auxiliary state operations (e.g., TodoWrite)*
22:              $\mathcal{E}_{t+1} \leftarrow \mathcal{E}_t$,   $g_{t+1} \leftarrow g_t$
23:          **end if**
24:      **end if**
25:      $\mathcal{H}_{t+1} \leftarrow \mathcal{H}_t \cup \{(a_t, o_t)\}$              ◁ *procedural trace always grows; captures full exploration history*
26:      $t \leftarrow t + 1$
27: **end while**
28: **Decoupled reasoning:** $y \sim P(y \mid q, \mathcal{E}_t)$         ◁ *final answer derived solely from $\mathcal{E}_t$; $\mathcal{H}_t$ is discarded to ensure trace-independent reasoning*
29: **return** $y$

---

## D. Dataset Details

**Evaluation Principles and Motivation.** To strictly evaluate the robustness of our approach, we employ two complementary benchmarks that scrutinize distinct dimensions of long-context understanding: **scale** and **reasoning depth**. First, we utilize ∞BENCH (Zhang et al., 2024) to assess the model's capacity to handle *ultra-long sequences* (averaging 200k tokens), testing the limits of effective context windows across diverse domains. Second, we incorporate **LOOGLE-V2** (He et al., 2025) to evaluate *long-dependency reasoning*, specifically targeting complex scenarios where evidence is non-contiguously scattered throughout the text. This combination ensures a comprehensive assessment, distinguishing between simple retrieval capabilities over long contexts and the ability to perform multi-hop reasoning and information aggregation. Due to computational constraints, we evaluate on a stratified random sample of 500 instances from LOOGLE-V2 and 1,000 instances from ∞BENCH; the same sampling is used consistently across all methods and ablations.

### D.1. ∞BENCH

**Overview:** ∞BENCH (Zhang et al., 2024) is the first large-scale benchmark designed to evaluate LLMs on context windows surpassing 100k tokens. The dataset features an average data length of approximately 200k tokens, with maximum sequence

lengths exceeding 1M tokens. Unlike benchmarks relying solely on synthetic extension, ∞BENCH comprises a diverse mix of synthetic and realistic tasks spanning English and Chinese languages.

**Task selection.**    We do not use all 12 tasks in ∞BENCH. We exclude `Math.Calc`, where nearly all methods (including frontier LLMs) achieve close to 0% accuracy, suggesting the task setup may not be well-suited for current evaluation. We also exclude certain open-ended generation tasks (e.g., summarization and dialogue) because the benchmark's automatic evaluation metrics for free-form outputs are not sufficiently reliable for fair cross-method comparison. The full suite of 12 tasks is categorized as follows:

**Retrieval Tasks:** These tasks evaluate the model's ability to locate specific information within a noisy context (needle-in-a-haystack).

- `Retrieve.PassKey`: The model must retrieve a specific 5-digit random number hidden within a large corpus of irrelevant text.

- `Retrieve.Number`: A higher-resolution retrieval task where the model must identify a specific number within a sequence containing successive repetitive digits, testing local attention capabilities.

- `Retrieve.KV`: Involves retrieving the value corresponding to a specific key within a massive, synthetically generated JSON object containing distinguishable and indistinguishable noise.

**Code Tasks:**

- `Code.Debug`: A realistic task derived from PyPI repositories. The model must locate a subtle, artificially injected bug (e.g., incorrect variable name or logic error) within a massive codebase (avg. 114.7k tokens).

- `Code.Run`: A synthetic task requiring the model to simulate the execution of a long sequence of simple Python functions (primarily arithmetic operations), testing long-term state tracking.

**Mathematical Reasoning:**

- `Math.Calc`: Requires the model to compute the final result of a long arithmetic expression containing addition and subtraction, testing sequential processing without information loss.

- `Math.Find`: The model is provided with a long array of numbers and must identify specific statistical elements, such as the median or the largest number.

**Long-Context Text Understanding (Novel & Dialogue):** These tasks are derived from real-world novels and scripts, requiring aggregation and reasoning over narrative elements.

- `En.Sum`: The model must generate a concise summary of a full-length English novel (avg. 103.5k tokens).

- `En.QA`: Open-ended Question Answering based on English novels. Questions often require aggregating scattered evidence or filtering specific details (e.g., How much money did character A spend in total?).

- `En.MC`: Multiple-choice questions based on English novels, designed to be challenging by requiring distinguishing between plausible distractors.

- `Zh.QA`: Open-ended Question Answering based on Chinese novels (avg. 2M tokens), testing cross-lingual long-context capabilities.

- `En.Dia`: Derived from movie and drama scripts. The model must identify the masked name of a character based on the dialogue history and context cues.

### D.2. LOOGLE-V2

**Overview:** LOOGLE-V2 (He et al., 2025) is a comprehensive benchmark designed to assess *true long-context understanding* rather than simple retrieval. It specifically targets the inability of current LLMs to process *Long-Dependency Tasks*, where the required evidence is interdependent and scattered throughout the text. The benchmark utilizes strictly real-world data sources (not synthetic) to minimize data contamination and ensure practical relevance.

**Data Statistics:** The dataset contains 1,934 QA instances with an average context length of approximately 250k tokens, ranging from 16k to over 2M tokens.

**Domain and Task Composition:** LOOGLE-V2 spans four specialized domains, each requiring distinct reasoning capabilities:

- **Finance:** Involves analyzing annual reports (10-K) spanning multiple years. Tasks include *Trend Analysis* (tracking financial metrics over time), *Metric Calculation*, and *Cross-Company Comparison*. Success requires aggregating quantitative data dispersed across hundreds of pages.

- **Law:** Utilizes US legal cases and articles. Tasks such as *Legal Case Retrieval* and *Legal Article Extraction* require the model to identify implicit fact patterns and analogous reasoning over extensive legal corpora.

- **Code:** Focuses on repository-level comprehension. Tasks include *Call Graph Analysis* (inferring multi-hop function dependencies) and *Version Control* (identifying code modifications between commits).

- **Game:** Analyzes logs from *Counter-Strike 2* and *Crafter*. Tasks like *User Behavior Analysis* require inferring global game states and strategy from sequential event logs.

**Reasoning Challenge:** Unlike standard retrieval benchmarks, LOOGLE-V2 tasks are formulated to prevent short-cut learning. The answer distribution relies on *inter-dependency*, meaning the model must jointly interpret multiple pieces of evidence (multi-hop reasoning) to derive the correct conclusion, rather than retrieving a single sentence.

## E. Experimental Details

### E.1. Evaluation Metrics: Cost and Token Efficiency

We clarify the precise definition of **Cost (k)** and **Token Efficiency (Token Eff.)** reported in Table 1 and throughout the experiments.

**Token Cost Definition.** For each query instance, we define the **token cost** as the total number of tokens processed by the LLM during inference, measured in thousands (k). Specifically:

$$\text{Cost} = \frac{1}{1000} \left( \sum_{t=0}^{T} \text{Input}_t + \sum_{t=0}^{T} \text{Output}_t \right) \tag{6}$$

where $T$ is the number of inference calls (turns) for that instance.

- **For monolithic LLMs (full-context):** A single inference call is made. $\text{Input}_0$ includes the full document (or truncated context) plus the query prompt; $\text{Output}_0$ is the generated answer. Thus $\text{Cost} = (\text{Input}_0 + \text{Output}_0)/1000$.

- **For agentic methods (SCOUT, ReadAgent, etc.):** Multiple turns of interaction occur. At each turn $t$:
  - $\text{Input}_t$ includes the system prompt, query, interaction history $\mathcal{H}_t$ (or a summary thereof), epistemic state $\mathcal{E}_t$ (for SCOUT), and any tool-returned content (observations, file content, search results).
  - $\text{Output}_t$ includes the model's reasoning and the generated action (tool call or final answer).

  The total cost is the **cumulative sum** over all turns until termination.

**Aggregation Across Instances.** For each benchmark, we report the **mean** cost across all evaluated instances. That is, if there are $N$ instances with individual costs $\{c_1, \ldots, c_N\}$, we report $\bar{c} = \frac{1}{N} \sum_{i=1}^{N} c_i$.

**Multiple Runs.**    All main experimental results reported in Table 1 are averaged over **three independent runs** with different random seeds. This ensures the stability and reproducibility of the reported numbers, particularly for agentic methods where stochastic sampling in the policy may introduce variance across runs.

**Token Efficiency Definition.**    Token Efficiency measures how much accuracy is achieved per unit of token cost:

$$\text{Token Eff.} = \frac{\text{Accuracy (\%)}}{\text{Cost (k)}} \tag{7}$$

A higher Token Eff. indicates that the method extracts more correct answers per thousand tokens processed, reflecting better information extraction efficiency. This metric is particularly informative for comparing methods with different accuracy–cost trade-offs.

### E.2. Baseline Implementation

We provide implementation details for baseline methods to ensure reproducibility.

**Closed-Source LLMs.**    We evaluate closed-source models via their official APIs. Table 6 summarizes the maximum input context length for each model used in our experiments.

*Table 6.* **Closed-source models used in experiments.** We report the maximum input context length, API provider, and relevant notes for each model.

| Model | Max Context | Provider | Notes |
|---|---|---|---|
| GPT-4.1 | 1M | OpenAI | Extended context variant |
| GPT-5.1-chat | 128k | OpenAI | Standard chat endpoint |
| Gemini-2.5-Pro | 1M | Google | Long-context flagship |
| Gemini-3-Pro | 1M | Google | Latest generation |
| Claude-Sonnet-4.5 | 200k | Anthropic | SCOUT backbone |

**Truncation Strategy.**    For documents exceeding a model's maximum context length, we apply **middle truncation**: we preserve the first $\lfloor L_{\max}/2 \rfloor$ tokens and the last $\lfloor L_{\max}/2 \rfloor$ tokens, discarding the middle portion. In Python notation, this corresponds to `tokens[:max_len//2] + tokens[-max_len//2:]`. This strategy preserves both the beginning (which often contains structural information, headers, or introductions) and the end (which may contain conclusions, summaries, or answers), following the standard practice in LTU benchmark evaluation.

**Open-Weight LLMs.**    For open-weight model evaluation, we use **Qwen2.5-Instruct-72B** with an extended context window. The model originally supports a 32k context window; we extend it to 128k using the YaRN (Yet another RoPE extensioN) technique with the following `rope_scaling` configuration: `factor=4.0`, `original_max_position_embeddings=32768`, `type="yarn"`. The same middle-truncation strategy is applied for documents exceeding 128k tokens.

**Claude Code.**    We use the official **Claude Code SDK** (Anthropic's free agent framework) to implement the Claude Code baseline. For each long-text evaluation instance, we save the document as a `.txt` file in the local filesystem and provide the agent with the file path along with the query. The agent is given free access to file-reading tools (`TodoWrite`, etc.) and can autonomously decide which portions of the document to read, search, or navigate. This setup is **identical to our SCOUT evaluation protocol**: both methods receive the document as a file rather than in-context, ensuring a fair comparison of agentic exploration strategies. The agent may invoke multiple tool calls to progressively gather information before producing a final answer.

**Other Baselines.**    For ReadAgent, GraphReader, and MemAgent, we follow their official implementations and hyperparameter settings as reported in their respective papers. All agentic baselines in the proprietary model comparison (Table 1) use Claude-Sonnet-4.5 as the backbone to ensure fair comparison.

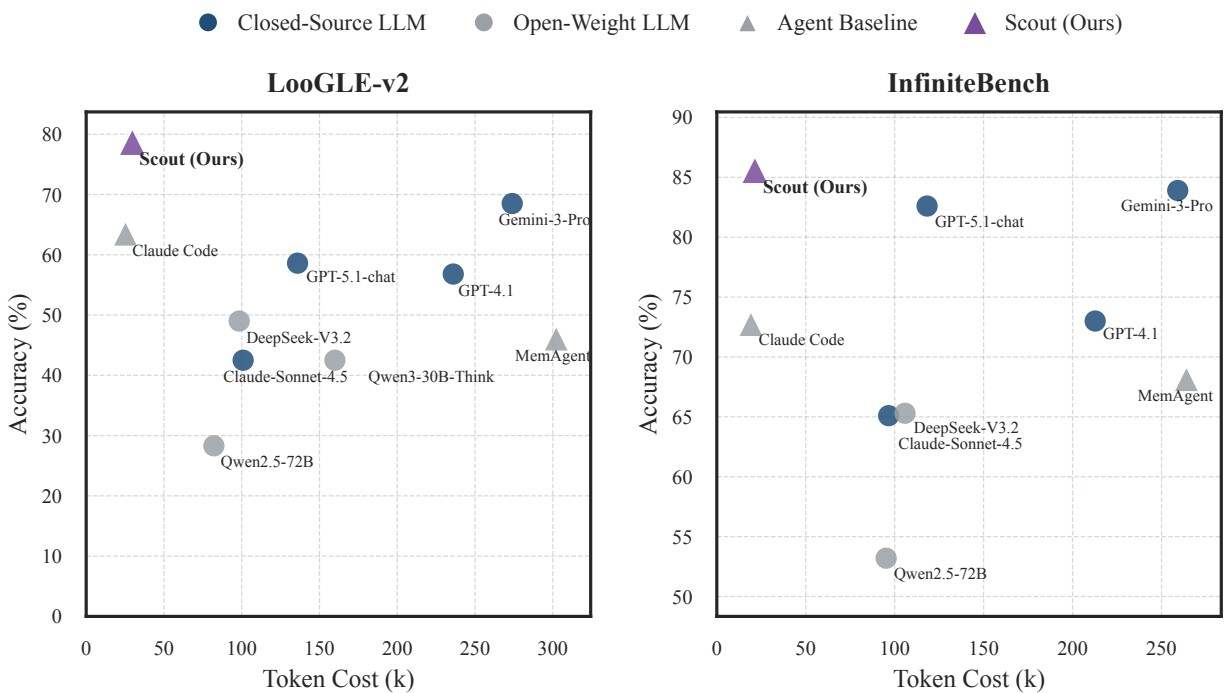

*Figure 4.* **Comprehensive accuracy–cost tradeoff. Left**: LOOGLE-V2. **Right**: ∞BENCH. We compare closed-source LLMs (blue circles), open-weight LLMs (gray circles), agent baselines (gray triangles), and SCOUT (purple triangle). SCOUT achieves Pareto-optimal performance on both benchmarks, combining the highest accuracy with low token cost. Notably, open-weight LLMs generally underperform closed-source counterparts, yet SCOUT enables competitive performance even with open-weight backbones (see Section 3.5).

### E.3. Training Details for Open-Weight Models

This section provides implementation details for the open-weight post-training experiments reported in Section 3.5.

**SFT Data Construction.** We construct SFT data strictly from datasets that are *disjoint* from our evaluation benchmarks to prevent any test contamination. Our training sources include LONGBENCH-V2 (Bai et al., 2025) and NARRATIVEQA (Kociský et al., 2018). For the **LLM setting**, we directly fine-tune on the original (query, long-context, answer) triples. For the **SCOUT setting**, we use Claude-Sonnet-4.5 solely as a high-accuracy trajectory generator to obtain interaction traces, then apply a multi-stage filtering pipeline:

1. **Correctness filtering**: retain only trajectories that produce a correct final answer.

2. **Behavioral pattern filtering**: among correct trajectories, keep those whose interaction patterns satisfy all three structural requirements (R1–R3 in Section 2.2): relevance alignment, progress awareness, and sufficiency control.

3. **Shortest rejection sampling**: when multiple trajectories remain for the same instance, select the shortest one to encourage efficient foraging behavior.

4. **Deduplication**: remove near-duplicate trajectories to ensure diversity in the training set.

The resulting set of filtered trajectories forms the SCOUT SFT training data. Overall, the SFT training data is fully independent from the test data, ensuring no leakage into evaluation.

**Budget-Matched Training Protocol.** To ensure fair comparison between the monolithic LLM setting and the SCOUT agent setting, we use the same optimization recipe (full fine-tuning for two epochs) and report the resulting data scale and average sequence lengths for each regime.

- **LLM SFT**: 500 training instances with an average sequence length of ∼80k tokens; we train for 2 epochs.

- **SCOUT SFT**: 1,200 successful trajectories with an average trajectory length of ∼30k tokens; we train for 2 epochs.

- **Token Counting**: We define training tokens as the sum of input context tokens and generated tokens consumed during optimization. For DAPO, this includes all rollout tokens used to compute policy gradients.

**DAPO Training Setup.** We implement DAPO training with the **veRL** (Sheng et al., 2024) framework using a GRPO-style advantage estimator and DAPO reward management. Unless otherwise stated, we keep the policy architecture and tool interface identical to SFT and optimize the same backbone with full-parameter updates. Key settings are:

- **Sampling / rollouts**: stochastic decoding with temperature 1.2; $n=8$ rollouts per prompt; generate candidates with batch size 128 and train on a filtered batch size 64 (up to 10 generation batches for filtering).

- **Filtering**: enable group filtering with metric `seq_reward` to retain higher-quality trajectories for policy updates. We additionally apply **learnability filtering**: for each prompt, we require that the success rate across rollouts falls within 25%–75%, discarding prompts that are either too easy (already solved by most rollouts) or too hard (solved by almost none), so that policy gradients concentrate on instances with meaningful learning signal.

- **IS ratio clipping**: asymmetric clip ratios with lower/upper bounds 0.8 and 1.28 (i.e., clip_ratio_low=0.2, clip_ratio_high=0.28), and dual-clip constant $c = 10.0$.

- **Length constraints / shaping**: max prompt length 8192 tokens and max response length 65536 tokens; we filter overlong prompts and apply an overlong buffer penalty (buffer length 4096, penalty factor 1.0).

- **Optimization**: AdamW with learning rate $1 \times 10^{-6}$; token-mean loss aggregation; no KL loss (use_kl_loss=false).

- **Training schedule / compute**: 4 epochs for 100 total training steps on a 16-node × 8-GPU cluster (128× NVIDIA H20). Each node has 192 physical CPU cores (384 logical cores) and runs Linux 5.4 (x86_64).

**Hyperparameters.** Unless otherwise noted, we use the same SFT hyperparameters for both regimes: AdamW optimizer, learning rate $5 \times 10^{-6}$, per-device batch size 1 with gradient accumulation 2, gradient checkpointing enabled, warmup ratio 0.05, and full-parameter fine-tuning (train_type = full). We do not hold out a validation split (split_dataset_ratio = 0) and log every step (logging_steps = 1).

# F. Additional Experimental Results

This section provides supplementary experimental results that complement the main findings in Section 3.

## F.1. Results with Alternative Backbones

The main results in Section 3.2 use Claude-Sonnet-4.5 as the unified backbone for all agentic methods. Here we report additional experiments with alternative frontier backbones to demonstrate that SCOUT's gains generalize across different LLM capabilities.

**Claude-Sonnet-3.7.** Table 7 presents results using Claude-Sonnet-3.7 as the backbone.

*Table 7.* **SCOUT with Claude-Sonnet-3.7 backbone.** Comparison against baselines on LOOGLE-V2 and ∞BENCH.

| Method | LOOGLE-V2 | | ∞BENCH | |
|---|---|---|---|---|
| | Acc (%) | Cost (k) | Acc (%) | Cost (k) |
| Claude-Sonnet-3.7 (Full Context) | 49.4 | 99.2 | 76.1 | 120.6 |
| ReadAgent | 20.1 | 304.2 | 36.1 | 397.8 |
| GraphReader | 32.4 | 406.0 | 41.9 | 411.5 |
| MemAgent | 51.3 | 306.0 | 75.3 | 218.6 |
| **SCOUT** | **65.4** | **36.4** | **85.1** | **16.7** |

*Table 8.* **SCOUT with Claude-Sonnet-4 backbone.** Comparison against baselines on LOOGLE-V2 and ∞BENCH.

| Method | LOOGLE-V2 | | ∞BENCH | |
|---|---|---|---|---|
| | Acc (%) | Cost (k) | Acc (%) | Cost (k) |
| Claude-Sonnet-4 (Full Context) | 46.0 | 101.3 | 69.0 | 121.5 |
| ReadAgent | 22.7 | 304.2 | 32.1 | 357.8 |
| GraphReader | 33.5 | 411.6 | 44.4 | 412.9 |
| MemAgent | 52.9 | 360.8 | 70.1 | 224.4 |
| **SCOUT** | **68.4** | **26.2** | **77.6** | **17.7** |

*Table 9.* **Open-weight LLM results on LOOGLE-V2 and ∞BENCH.** Full-context evaluation with Acc (↑, %) and Cost (k) ↓.

| Model | LOOGLE-V2 | | ∞BENCH | |
|---|---|---|---|---|
| | Acc | Cost (k) | Acc | Cost (k) |
| Qwen2.5-Instruct-32B-128k (Qwen Team, 2024) | 29.3 | 83.5 | 49.2 | 95.4 |
| Qwen2.5-Instruct-72B-128k | 28.3 | 82.1 | 53.2 | 95.2 |
| Qwen3-30B-A3B-Instruct (Yang et al., 2025) | 41.8 | 151.6 | 55.1 | 133.4 |
| Qwen3-30B-A3B-Thinking | 42.5 | 159.9 | 62.8 | 133.6 |
| DeepSeek-V3.2 (DeepSeek-AI, 2025) | 49.0 | 98.4 | 65.3 | 105.8 |
| Qwen3-235B-A22B-Thinking | 30.3 | 164.0 | 59.6 | 134.4 |

**Claude-Sonnet-4.** Table 8 presents results using Claude-Sonnet-4 as the backbone.

### F.2. Open-Weight Model Results

Table 9 presents the performance of open-weight LLMs on LOOGLE-V2 and ∞BENCH under full-context evaluation. We evaluate a range of models including Qwen2.5-Instruct (Qwen Team, 2024), Qwen3 (Yang et al., 2025), and DeepSeek-V3 (DeepSeek-AI, 2025).

Figure 4 visualizes the accuracy–cost tradeoff across all evaluated models on both benchmarks. LLM methods are shown as circles (blue for closed-source, gray for open-weight), while agent methods are shown as triangles (gray for baselines, purple for SCOUT). SCOUT achieves Pareto-optimal performance, attaining the highest accuracy at substantially lower token cost.

### F.3. Comparison with Retrieval-Augmented Generation Baselines

We compare SCOUT against retrieval-augmented generation (RAG) baselines to evaluate the effectiveness of our active foraging paradigm relative to standard retrieval-based approaches.

**MemAgent (RL-14B).** In addition to evaluating MemAgent with Claude-Sonnet-4.5 as the backbone (reported in the main results), we also test **RL-MemoryAgent-14B** (Yu et al., 2025a), a 14B parameter model trained with multi-turn RL for long-context memory-augmented reasoning. This open-weight variant represents a specialized small model trained specifically for memory-based document understanding. Results are reported in Table 11 alongside other RAG and agentic baselines.

**Analysis.** As shown in Table 11, SCOUT achieves the highest accuracy on both benchmarks while maintaining low token cost. MemAgent (RL-14B), a specialized 14B model trained with multi-turn RL, achieves moderate accuracy at lower cost but falls substantially short of SCOUT in accuracy. The results suggest that SCOUT's active foraging approach, which iteratively explores and refines its epistemic state, is more effective for long-dependency reasoning than segment-based reading or memory-augmented strategies.

*Table 10.* **Paradigm ablation on** $\infty$**BENCH.** Removing $\mathcal{E}_t$ (ReAct-style) degrades long-horizon retention; removing $\mathcal{A}_{\text{forage}}$ collapses performance by preventing access to $\mathcal{D}$. Grounding and auxiliary file tools provide additional gains.

| Variant | Acc (%) | Δ | Cost (k) |
|---|---|---|---|
| **SCOUT (Full)** | **85.6** | – | **21.4** |
| w/o $\mathcal{E}_t$ (ReAct-style) | 75.5 | -10.1 | 22.5 |
| w/o $\mathcal{A}_{\text{forage}}$ (no access to $\mathcal{D}$) | 25.4 | -60.2 | 22.3 |
| w/o File Tools | 77.0 | -8.6 | 24.1 |
| w/o Grounding ($\alpha$) | 81.3 | -4.3 | 19.6 |

*Table 11.* **Comparison with RAG and agentic baselines on LOOGLE-v2 and** $\infty$**BENCH.** We report accuracy (%) and token cost (k tokens).

| Method | LOOGLE-v2 | | $\infty$BENCH | |
|---|---|---|---|---|
| | Acc (↑, %) | Cost (k) ↓ | Acc (↑, %) | Cost (k) ↓ |
| ReadAgent (Lee et al., 2024) | 27.7 | 302.6 | 42.3 | 481.8 |
| GraphReader (Li et al., 2024b) | 31.6 | 408.1 | 43.1 | 327.4 |
| MemAgent (Yu et al., 2025a) | 46.0 | 302.2 | 68.1 | 264.0 |
| MemAgent (RL-14B) (Yu et al., 2025a) | 36.3 | 60.4 | 50.6 | 78.5 |
| Claude Code (Anthropic, 2025b) | 63.4 | **25.3** | 72.7 | **19.2** |
| **SCOUT** | **78.7** | 29.7 | **85.6** | 21.4 |

## F.4. Latency Analysis

While the main text focuses on token cost as the primary efficiency metric, we additionally report wall-clock latency to provide a more complete picture. Note that wall-clock latency depends on many factors beyond the method itself, including hardware configuration, network conditions, API provider throttling and batching behavior, and serving concurrency. All measurements below were conducted under a unified environment with the same API endpoints; however, the absolute values should be interpreted as *relative comparisons* rather than definitive benchmarks.

**Overall latency.** Table 12 compares latency on LOOGLE-v2. For agentic methods, latency is the cumulative wall-clock time across all turns; for LLMs, it is the end-to-end response time of a single API call.

*Table 12.* **Latency comparison on LOOGLE-v2.** [†] 1M input token limit. [‡] 200K input token limit (middle truncation applied for longer documents).

| Method | Category | Latency (s) |
|---|---|---|
| SCOUT | Ours | 153.9 |
| MemAgent | Agent | 212.4 |
| Claude-Sonnet-4.5 | LLM | 28.8[‡] |
| Gemini-3-Pro | LLM | 57.2[†] |

**Latency scaling with context length.** Table 13 shows how latency changes as context length increases from 64K to 1M+ tokens. SCOUT maintains **near-constant latency** across the entire range, while monolithic LLMs see latency grow with input length, and MemAgent's latency increases significantly at longer contexts.

**Analysis.** SCOUT has higher absolute latency than monolithic LLMs due to multi-turn interaction overhead, but is comparable to or better than MemAgent. Importantly, SCOUT's near-constant latency scaling contrasts with MemAgent (whose latency grows from 106s to 559s) and monolithic LLMs (whose latency grows with input length up to their context limit). Furthermore, SCOUT's substantially lower token consumption ($\sim$1/8 of monolithic methods) enables higher serving concurrency under the same compute budget, partially offsetting its per-query latency overhead. We discuss latency optimization as an important direction for future work in Appendix A.

*Table 13.* **Latency scaling with context length on LOOGLE-V2 (seconds).** † 1M input token limit. ‡ 200K input token limit.

| Context | SCOUT | MemAgent | Gemini-3-Pro[†] | Claude-4.5[‡] |
|---|---|---|---|---|
| 64K | 150.0 | 106.3 | 34.5 | 15.1 |
| 128K | 154.2 | 118.5 | 43.4 | 25.3 |
| 256K | 154.9 | 177.3 | 58.9 | 36.8 |
| 512K | 157.8 | 310.8 | 84.3 | 36.5 |
| 1M+ | 153.2 | 559.3 | 91.7 | 37.1 |

### F.5. Ablation Studies on ∞BENCH

We provide complete ablation results on ∞BENCH, complementing the LOOGLE-V2 results in the main text.

**Paradigm Ablation.** Table 10 evaluates the contribution of each architectural component in the SCOUT paradigm, mirroring the LOOGLE-V2 ablation in the main text (Table 2).

### F.6. Analysis of Training Gains under SCOUT

**Why SCOUT unlocks large gains while LLM-only SFT does not.** LLM-only SFT continues to optimize a *monolithic history-conditioned* behavior, where a single growing interaction log must jointly support exploration control and final reasoning. In long-horizon agent settings, this design is known to be fragile under context-budget pressure, because extended search cycles rapidly inflate the history and dilute answer-relevant signal, leading to brittle and inefficient reasoning (Wu et al., 2025). A natural workaround is context compression, but recent agent-focused studies caution that naive truncation or generic summarization can discard answer-critical details, creating a length–fidelity tradeoff rather than a free win (Kang et al., 2025). Practical agent systems therefore emphasize *structured* context management that preserves key decisions and unresolved issues while discarding redundant tool outputs, highlighting that *how* information is retained matters as much as *how much* is retained.

Crucially, LLM-only SFT provides no explicit learning signal for *stateful tool semantics* and long-horizon control, such as when to commit stable knowledge, how to react to diagnosed missing information, or when to terminate once query-sufficiency is reached. This connects to a broader observation in multi-turn and long-horizon agent training: trajectory-level credit assignment is non-trivial, and policy optimization can fail to attribute improvements to the right steps without an interaction-aligned structure (Li et al., 2025a). By contrast, post-training under SCOUT matches the target interaction protocol and exposes an explicit state/action interface, so lightweight adaptation can directly optimize the intended control semantics. This decoupling makes learning better conditioned and more sample-efficient, while DAPO further strengthens long-horizon behavior by optimizing full trajectories under the same state/action regime. Incorporating uncertainty-aware preference optimization (Wang et al., 2025) to handle noisy trajectory-level feedback is a promising future direction.

## G. Qualitative Trajectory Examples

To make SCOUT's behavior concrete, we present representative trajectories that highlight (i) coarse-to-fine foraging, (ii) knowledge distillation into provenance-aware epistemic units, and (iii) the progressive refinement and convergence of the Epistemic State $\mathcal{E}_t$. Unlike examples that reduce to keyword matching, the case below is a long-dependency document understanding task over a technical paper ($\sim$200k tokens): the agent must integrate evidence scattered across system descriptions, privacy controls, and implementation constraints to resolve a multi-step inference query about dependency propagation under a permission change.

---
**SCOUT Trajectory: Long-Dependency Multi-Hop Reasoning (19 steps)**

**[Query]** Document: ContextCam technical paper (221k tokens, CHI '24)
**[Query]** Question: If I turn off the permission to obtain location information, what context information can ContextCam extract *at most*?
**[Query]** Options: (A) facial expression, location, music, screen content, weather    (B) facial expression, music, screen content, weather    (C) facial expression, music, screen content    (D) facial expression, screen content

---

**[Action]** $t$=1: `GetFileInfo(context_file.txt)`
**[Obs]** *file_size: 98KB, estimated_tokens: 221,320, needs_normalization: false*

**[Think]** The file is long (220k tokens). I should avoid full reading: first locate relevant sections with cheap searches, then read only the small windows needed to verify dependencies and permissions.

**[Action]** $t$=2: `TodoWrite("Plan: (i) identify all supported context types; (ii) find detector descriptions (Table 4) to derive dependencies; (iii) locate privacy/permission policy describing what happens when location permission is off; (iv) verify there is no fallback or indirect inference for location/weather; (v) conclude max extractable types and map to option.")`

---

**[Action]** $t$=3: `Grep("five types of contextual information|contextual information", case_insensitive=true)`
**[Obs]** *Line 437 → "...five types of contextual information: location, facial expression, music, screen content, and weather..."*
*Line 438 → "We focus on these context types..."*
*Line 512 → "...contextual information includes ..." ...*

**[Update $\mathcal{E}$]** Commit $e_1$: ContextCam supports **5 context types**: {Location, Screen Content, Facial Expression, Weather, Music}.
[Anchor: Lines 437–438]

**[Action]** $t$=4: `Grep("Table 4|Context detectors used|Weather:|Location:", case_insensitive=true)`
**[Obs]** *Line 1810 → "Table 4: Context detectors used in the implemented system."*
*Line 1813 → "Location: GPS ... network positioning ..."*
*Line 1821 → "Screen Content: Capturing the text displayed on the user's current screen ..."*
*Line 1825 → "Facial Expression: ...capturing the user's frontal face ..."*
*Line 1833 → "Weather: Retrieving real-time weather data **for the user's location** ..."*
*Line 1837 → "Music: If music is detected in the environment, returning its song title ..."*
*Line 1842 → "...detectors feed into the Context Selector ..." ...*

**[Update $\mathcal{E}$]** Commit $e_2$ (Detector descriptions from Grep):
Location uses **GPS + network positioning**.
Weather is retrieved **for the user's location**.
Screen/Face/Music rely on screen/camera/mic signals.
[Anchor: Table 4, Lines 1810–1842]

**[Evaluate]** $t$=6: `Evaluate(`$\mathcal{E}_t$`, query)`
**[Obs]** *is_sufficient: false*
*missing_info:*
*(1) Explicit policy: what the system does when **Location permission is off**.*
*(2) Explicit dependency: does Weather **consume Location output** or can it run independently?*
*(3) "At most" check: any **fallback** (IP/manual city) or **indirect inference** that could still provide Weather/Location.*

**[Think]** Need long-range evidence across privacy controls + architecture + implementation limitations.

---

**[Action]** $t$=7: `Grep("disable OS-level|Location permission|indirectly infer location|dependency-based", case_insensitive=true)`
**[Obs]** *Line 5519 → "Users can disable OS-level **Location permission** ..."*
*Line 5527 → "When disabled, the **Location detector is not executed**."*
*Line 5536 → "Detectors that require Location output are **automatically disabled** ..."*
*Line 5576 → "...we avoid inferring location indirectly from screen/app content..."*
*Line 5582 → "...privacy guarantee ..." ...*

**[Update $\mathcal{E}$]** Commit $e_3$ (Policy rule from Grep): If Location permission is off: Location detector not executed; dependent detectors disabled; indirect location inference is forbidden.
[Anchor: Lines 5519, 5527, 5536, 5576]

**[Action]** $t$=9: `Grep("Weather takes as input the Location detector|Weather is marked N/A", case_insensitive=true)`
**[Obs]** *No matches found.*

**[Think]** Direct phrasing may differ. Locate the system overview/pipeline section, then read a small window to confirm the dependency.

**[Action]** $t$=10: `Grep("System overview|pipeline|Sensor Layer|Context Selector", case_insensitive=true)`
**[Obs]** *Line 1212 → "System Overview"*
*Line 1219 → "Sensor layer → detectors → context selector"*
*Line 1226 → "...detector outputs are aggregated ..." ...*
*Line 1230 → "Weather detector ... input ..." (truncated)*

**[Action]** $t$=11: `Read(offset=1216, limit=75)`
**[Obs]** *System overview excerpt (summarized):*
– *"Sensor Layer → Detectors → Context Selector"*
– *"Weather detector takes as input the **Location detector's output** (lat/lon or coarse city)."*
– *"If Location is unavailable, Weather is marked `N/A` and omitted from the context bundle." ...*
**[Update $\mathcal{E}$]** Commit $e_4$ (Dependency proof): Weather consumes Location output; if Location unavailable, Weather is omitted.
[Anchor: Lines 1216–1238]

**[Action]** $t$=13: `Grep("IP-based geolocation|manual city entry|default city", case_insensitive=true)`
**[Obs]** *No matches found.*

**[Think]** Fallback wording may appear only in limitations/prototype discussion. Read the limitations window around where fallback is discussed.

**[Action]** $t$=14: `Read(offset=6346, limit=55)`
**[Obs]** *Implementation/limitations excerpt (summarized):*
– *"No fallback to **IP-based geolocation**."*
– *"No manual city entry in current prototype."*
– *"If Location permission is denied, location-dependent detectors are skipped." ...*
**[Update $\mathcal{E}$]** Commit $e_5$ (No-fallback evidence): Prototype provides no IP/manual fallback; location-dependent detectors are skipped when permission denied.
[Anchor: Lines 6350–6368]

**[Action]** $t$=16: `Grep("Screen Content: Capturing|Facial Expression:|Music: If music", case_insensitive=true)`
**[Obs]** *Line 1821 → "Screen Content: Capturing the text displayed on the user's current screen ..."*
*Line 1825 → "Facial Expression: ...capturing the user's frontal face ..."*
*Line 1837 → "Music: If music is detected in the environment, returning its song title ..." ...*
**[Update $\mathcal{E}$]** Commit $e_6$ (Independence evidence): Screen/Face/Music detectors operate on screen/camera/mic signals and do not require Location output.
[Anchor: Lines 1821, 1825, 1837]

---

**[Update $\mathcal{E}$]** Commit $e_7$ (Synthesis):
From $e_3$–$e_5$, if Location permission is off, Location is unavailable and any Location-dependent detector is disabled, with no fallback or indirect inference.
From $e_4$, Weather is Location-dependent and omitted when Location is unavailable.
Thus the *maximum* extractable context types are {Screen Content, Facial Expression, Music}.

**[Evaluate]** $t$=19: `Evaluate(`$\mathcal{E}_t$`, query)`
**[Obs]** *is_sufficient: **true**, confidence: 0.96*
*reasoning: "Policy disables Location and cascades to Weather; privacy guarantee forbids indirect inference; no fallback exists; remaining independent contexts are Screen, Face, Music."*

---

**Final Epistemic State $\mathcal{E}_{19}$:**
$e_1$: Supported context types = {Location, Screen, Face, Weather, Music}.
$e_2$: Detector lines (Table 4): Location uses GPS/network; Weather retrieved for user's location; others use screen/camera/mic.
$e_3$: Permission gating: Location permission off ⇒ Location detector not executed; dependent detectors disabled; no indirect location inference.
$e_4$: Dependency: Weather consumes Location output; if Location unavailable, Weather omitted.
$e_5$: Implementation: no IP/manual-city fallback; location-dependent detectors skipped when permission denied.
$e_6$: Evidence that Screen/Face/Music are Location-independent.
$e_7$: Synthesized conclusion: max extractable = {Screen, Face, Music}.

**[Answer]** (Reasoning from $\mathcal{E}_{19}$ only)
Location permission off ⇒ Location ×. Weather × (depends on Location; no fallback; no indirect inference).
Remaining: Facial Expression ✓, Music ✓, Screen Content ✓.
Answer: **(C) facial expression, music, and screen content**.

**Key observations (structure-aware long-dependency reasoning).**

- **Not a retrieval-only task, but long-dependency multi-hop reasoning.** Although the document is searchable, the query cannot be answered by a single keyword hit. The "at most" semantics requires *composing* multiple constraints that are distributed across distant sections and expressed in different forms: (i) the *definition space* (which context types exist), (ii) the *mechanism space* (what each detector consumes/produces, e.g., Table 4), (iii) the *policy space* (what

happens when Location permission is disabled), and (iv) the *implementation/limitations space* (whether any fallback or indirect inference is allowed). The final answer emerges only after chaining these heterogeneous pieces into a coherent implication graph: *permission-off $\Rightarrow$ Location unavailable $\Rightarrow$ Weather disabled $\wedge$ no fallback/indirect inference*.

- **Coarse-to-fine foraging guided by explicit insufficiency diagnosis.** SCOUT begins with low-cost global localization (`Grep` for context-type definitions and detector catalogs), and only performs fine local reading (`Read` in narrow windows) when `Evaluate` exposes missing proofs *or* when direct searches fail to match. This matches the design goal: defer expensive reading until the agent has a concrete hypothesis about *what* is missing and *where* it likely resides.

- **`Evaluate` serves as gap diagnosis, not a stop heuristic.** The first evaluation does not merely decide to continue; it produces a structured `missing_info` set that approximates $\mathcal{F}_q^\star \setminus \mathcal{E}_t$ (permission-off behavior, dependency direction, and fallback/indirect-inference exclusions). This diagnostic output directly determines the subsequent foraging targets (privacy controls $\to$ system overview $\to$ implementation limitations), making the control flow *query-adaptive* rather than task-agnostic.

- **Epistemic-state convergence toward a compact sufficient set.** Early $\mathcal{E}_t$ contains only coarse facts (supported context types and detector descriptions), which is insufficient. Each subsequent `Update` adds one *distilled* epistemic unit (policy rule, dependency proof, no-fallback guarantee, independence evidence), shrinking the gap to $\mathcal{F}_q^\star$ while preventing uncontrolled state growth. When `Evaluate` flips to `is_sufficient=true`, $\mathcal{E}_{19}$ has converged to a query-sufficient set that is close to $\mathcal{F}_q^\star$: sufficient for answering, yet nearly non-redundant.

- **State decoupling enables trace-independent, auditable reasoning.** Procedural details (tool calls, intermediate hits, failed matches, navigation) remain in the action history, while the final decision is derived solely from $\mathcal{E}_{19}$. Importantly, `Update` commits *knowledge* rather than raw observations (e.g., "Weather depends on Location" and "no fallback/indirect inference"), yielding a compact proof object with clear provenance anchors. This decoupling aligns with the paper's design: robustness to long interaction histories and improved auditability.

