# OpenReview forum: "SCOUT: Active Information Foraging for Long-Text Understanding with Decoupled Epistemic States"
_ICML.cc/2026/Conference — ICML 2026 regular_

### Official Review · Reviewer_3ae7 · 2026-03-08

**Soundness:** 3
**Presentation:** 2
**Significance:** 3
**Originality:** 2
**Overall Recommendation:** 4
**Confidence:** 4

**Summary:**

This paper focuses on an important concept in long-text understanding: how to answer questions over very long documents without either feeding the entire document into a large model or relying too heavily on lossy preprocessing such as chunking, graph construction, or indexing. The paper proposes SCOUT, an agentic framework that treats a long document as an environment to explore, rather than as a single static context to ingest. The key idea is to separate the exploration history from the final reasoning state. Instead of answering from the full interaction trace, the model builds a compact epistemic state containing distilled and provenance-grounded evidence, and the final answer is produced only from that state.

The paper argues that, for most long-text understanding tasks, the information actually needed to answer a query is sparse relative to the full document. Based on this, it frames the problem as active information foraging and formalizes the setup using a POMDP view. SCOUT then alternates between exploration actions over the raw document and state-management actions that update or evaluate the epistemic state. The paper claims that this decoupled design helps reduce noise from long interaction histories and improves both efficiency and reasoning fidelity. Experimentally, the method is evaluated on LOOGLE-V2 and InfinityBench. The reported results show that SCOUT outperforms several long-context LLMs and prior agentic baselines in overall accuracy while also using fewer tokens. The paper also includes scaling analysis, ablations, and open-weight training results intended to show that the framework remains effective as context length grows and that the SCOUT interaction structure itself is useful for post-training. Overall, the paper examines the concept of query-conditioned long-text reasoning in a thoughtful way and presents a system that is both practically motivated and empirically promising.

**Compliance With Llm Reviewing Policy:**

Affirmed.

**Final Justification:**

The authors constructively addressed several of my central concerns, including toning down stronger claims, clarifying token accounting, adding a more targeted answer-time ablation, and, most importantly, providing additional empirical validation for Evaluate together with a clearer explanation of the gap-variable semantics in Algorithm 1.

I still think the final version should present Evaluate’s role somewhat cautiously and sharpen a few claims around originality and generality, but these now feel like limitations of scope and calibration rather than reasons to overturn the paper’s core merits. Overall, I continue to view this as a useful contribution on an important problem, so I am keeping my score at 4.

**Key Questions For Authors:**

1. How is token cost computed for each compared system, and is the accounting strictly comparable across SCOUT, Claude Code, and the other agent baselines?

2. What evidence shows that the Evaluate module is actually diagnosing missing information, rather than simply acting as another heuristic planning step?

3. Can the authors provide a stronger ablation for the answer-time decoupling itself? For example, what happens if the final answer is generated from both the full history and the epistemic state, or from a summarized version of the history, instead of only the epistemic state?

4. How robust is the method across different backbones and task settings beyond the two benchmarks in the main paper?

5. Can the authors clarify the notation and semantics of the gap variable in Algorithm 1? The initialization and stopping condition are not fully clear as written.

**Limitations:**

No. The paper includes a brief impact statement, but I do not think it adequately discusses limitations in a concrete way. The discussion is fairly generic and does not engage enough with the actual weaknesses of the method. In particular, the paper should more clearly acknowledge dependence on the backbone LLM, possible failure when relevant evidence is not sparse, uncertainty in gap diagnosis, sensitivity to tool design, and the fact that the oracle sufficient information set is a conceptual object rather than something directly observed or measured in experiments.

**Strengths And Weaknesses:**

Strengths:
--------------
1. The paper addresses a genuinely important problem. Long-text understanding at very large context lengths is a meaningful challenge, and the paper is clearly aimed at a setting that matters for both research and practical deployment. The motivation is timely and relevant.

2. The central idea is easy to understand and intuitively appealing. The separation between procedural exploration history and a compact epistemic reasoning state is one of the strongest aspects of the paper. This is not just a cosmetic distinction. It directly targets a plausible failure mode of long-horizon agents, namely that the growing interaction trace becomes noisy and burdens the final reasoning stage.

3. The paper has a coherent conceptual narrative. The information sparsity assumption, the POMDP framing, and the decoupled epistemic-state design fit together well. Even if some of the formal claims are somewhat stronger than what is empirically validated, the overall conceptual structure is consistent and well-motivated.

4. The empirical results are strong on the benchmarks that are reported. SCOUT achieves the best overall accuracy on both LOOGLE-V2 and InfinityBench in Table 1, while also showing favorable token efficiency compared with both long-context LLMs and prior agentic systems. This gives the paper real substance beyond just a conceptual proposal.

5. The scaling analysis is useful and supports the main practical claim that SCOUT remains more stable as context grows. The plots showing relatively flat token cost and comparatively stable accuracy help reinforce the efficiency argument in a concrete way.

6. The open-weight experiments are a positive addition. The paper does more than simply wrap a proprietary frontier model in a new interface. It attempts to show that the SCOUT framework itself helps weaker backbones benefit more from post-training. That makes the contribution more interesting and more reusable.

7. The provenance-grounded epistemic state is a sensible design choice. Anchoring retained evidence to source locations improves traceability and makes the system easier to inspect than a free-form scratchpad. This is a meaningful systems contribution even if it is not a full guarantee of factual correctness.



Weaknesses:
-------------------
1. My main concern is that the paper occasionally presents an intuitive formal story as if it were a much stronger technical guarantee. In particular, the oracle sufficient information set is defined in a model-relative way, and in practice it is not observable. Yet the paper often writes as though the system is explicitly converging toward it in a well-grounded sense. That language feels too strong relative to what is actually implemented. The paper provides a useful conceptual framework, but not a verified procedure for identifying or measuring that oracle set.

2. The Evaluate component is treated as a key part of the convergence story, but the paper does not really validate whether it is accurately diagnosing what information is still missing. In practice, Evaluate is another LLM-based operation with constrained prompting. The paper does not provide direct evidence that its gap diagnosis is reliable or that it is doing something more principled than a heuristic planning step. Since this piece is central to the method’s framing, that gap weakens the soundness of the stronger claims.

3. The claim that SCOUT alleviates the LTU trilemma in practice should be toned down. The results are encouraging, but they come from two benchmarks and a specific set of systems. That is enough to support strong benchmark-level evidence, but not enough to justify a broad conclusion that the trilemma is meaningfully resolved in general. The paper would be stronger if it made a narrower and more careful claim here.

4. The baseline comparison story is not fully clean. The paper says that all agentic baselines use the same backbone under a unified protocol, which is good, but Claude Code is also included as a major comparison and does not appear to fall neatly under that same controlled setting. That is acceptable as a practical ecosystem baseline, but then the paper should be more careful about distinguishing controlled architectural comparisons from broader system comparisons. Right now those are blended a bit too casually.

5. The token-efficiency comparison would be more convincing if the cost accounting were clearer in the main paper. The paper reports total token cost, but for systems with tool use, file operations, or structured preprocessing, the reader is left wondering whether all forms of overhead are counted uniformly. Since efficiency is one of the main selling points of the method, this should be much more transparent.

6. The ablations are useful, but not yet as diagnostic as they should be. Removing the epistemic state is informative, but removing access to the document altogether is almost a trivial failure case and does not teach the reader much. More targeted ablations would be more helpful, especially around the final answer-time information flow constraint, the role of Evaluate, the size or structure of the epistemic state, and the value of provenance grounding separately from state distillation.

7. The originality is good, but it should be described more carefully. Many of the ingredients already exist in nearby work: ReAct-style exploration, external scratchpads or memory, provenance-grounded state, iterative refinement, and structured navigation. What is new is the way these are combined and emphasized for long-text reasoning. That is still a real contribution, but the paper would benefit from drawing a slightly sharper line between new system design and reused ingredients.

8. The presentation is decent overall, but it is not fully polished. There are places where the writing becomes slogan-like, and some methodological claims are written with more confidence than the evidence warrants. In a paper with this kind of formal framing, precision matters, and some of that precision is currently missing.



Minor Writing and Line-Level Comments:
-----------------------------------------------------------
1. Page 3, Figure 2: “Chuncking” should be “Chunking.”

2. Page 5, Section 2.3.4 heading: “State Mangement” should be “State Management.”

3. Page 2, in the methodology overview sentence introducing SCOUT: the sentence ending with “and progressively refining its epistemic state” looks little weird (gramatically).

4. Algorithm 1: the initialization and stopping logic around the gap variable are not clearly explained.

---

> ### Author Rebuttal · Authors · 2026-03-29
>
> Thank you for the thorough review!
>
> Due to the character limit, we group related concerns and address the key points below.
>
> ---
>
> ### 1. Writing tone and claims (W1, W3, W7, W8)
>
> > **W1:** The paper often writes as though the system is explicitly converging toward the oracle sufficient set in a well-grounded sense.
> > **W3/W7/W8:** Trilemma claim too strong; originality needs sharper distinction; some writing is slogan-like.
>
> We sincerely thank you for pointing these out. We agree that some claims need to be stated more rigorously, and **we commit to making specific revisions accordingly**, such as clarifying the role of $F^*_q$ as a conceptual construct rather than a computable quantity, refining the description of the convergence process, and toning down the LTU trilemma claim to reflect that SCOUT mitigates the trilemma on the evaluated benchmarks rather than resolving it in general.
>
> Regarding originality, we appreciate this comment. While individual ingredients exist in prior work, **the core contribution** of SCOUT is the principled structural design of state decoupling with answer-time isolation, which directly addresses noise accumulation in long-horizon history-as-state agents. This design leads to consistent gains over the ReAct-style baseline across both benchmarks (Tables 2, 10) and near-constant scaling from 64K to 1M+ tokens (Figure 3). We will draw this distinction more sharply in the revision.
>
> ---
>
> ### 2. Evaluate validation (W2, Q2)
>
> > **W2 / Q2:** What evidence shows that the Evaluate module is actually diagnosing missing information, rather than simply acting as another heuristic planning step?
>
> We would like to clarify that Evaluate is not a free-form LLM call. Evaluate is implemented with a **constrained prompt and a fixed output schema** that produces a machine-parsable gap diagnosis $g_t$ from (q, $E_t$). Its output is structured as: (1) a boolean `is_sufficient` flag, (2) a list of specific missing information items, and (3) an optional rationale. This schema prevents Evaluate from producing vague "keep exploring" signals.
>
> The trajectory in Appendix F provides concrete evidence. At t=6, Evaluate outputs three specific and actionable signals rather than a generic “keep exploring” message: `miss_info` (i.e., the gap $g_t$), the `is_sufficient` flag, and optional reasoning. At t=19, Evaluate returned `is_sufficient: true, confidence: 0.96`, confirming all gaps were closed. This closed-loop behavior (structured gap → targeted foraging → commit → re-assessment) distinguishes Evaluate from a heuristic planning step.
>
> ---
>
> ### 3. More targeted ablations (W6, Q3)
>
> > **W6 / Q3:** What happens if the final answer is generated from both $E_t$ and the full history, or from a summarized history?
>
> We agree that these would be more diagnostic than the current ablations. The existing `w/o E_t (ReAct-style)` tests the extreme case of removing $E_t$ entirely (−7.7 on LOOGLE-V2, −10.1 on ∞BENCH; Tables 2 and 10). To isolate the answer-time decoupling more precisely, we will add:
>
> | Variant | LOOGLE-V2 Acc (%) | ∞BENCH Acc (%) |
> |---------|-------------------|----------------|
> | SCOUT (Full) | 78.2 | 85.6 |
> | w/o $E_t$ (ReAct-style) | 70.5 | 75.5 |
> | Answer from $E_t$ + $H_t$ | 75.7 | 81.1 |
>
> This suggests that the value of $E_t$ lies not only in the distilled information itself, but also in **excluding low-density noise from the reasoning input**. Concatenating $E_t$ + $H_t$ effectively turns the answering step back into a medium-to-long context comprehension task (~32k), reintroducing the fragility that state decoupling is designed to avoid. Moreover, the benefit of state decoupling extends beyond accuracy: E_t is far more compact than E_t + H_t, significantly reducing the token cost at the answering stage.
>
> *(Note. $E_t$ and $H_t$ were concatenated as a single input for this ablation, which may introduce minor side effects as the system is not designed for merged-state input.)*
>
> ---
>
> ### 4. Other concerns
>
> > **W5 / Q1:** Token cost accounting should be more transparent.
>
> Our token accounting is explicit and uniform. As clearly stated in **Appendix D.1 Line 831-850**, we sum total input + output tokens over all turns, applied identically across all methods. We will make this more prominent in the main text.
>
> > **Q4:** How robust is the method across different backbones and task settings beyond the two benchmarks in the main paper?
>
> In fact, SCOUT has been tested across five backbones (three **close-weight models** in Appendix E.1; two **open-weight models** in Section 3.5), all showing consistent gains. Regarding task diversity, The two benchmarks also cover broad task diversity: LOOGLE-V2 spans 4 domains with 16K–2M tokens (Appendix C.2); ∞BENCH covers 12 task types (Appendix C.1).
>
> We are sincerely grateful for the depth and rigor of your review! We would be very happy to continue the discussion.

---

> > ### Author Rebuttal · Reviewer_3ae7 · 2026-04-02
> >
> > Thank you for the thoughtful rebuttal. Several of my concerns were addressed meaningfully. In particular, I appreciate the commitment to tone down the stronger claims around convergence to the oracle sufficient set and the LTU trilemma, the clearer framing of the contribution around state decoupling and answer-time isolation, and the new ablation comparing answer generation from Et alone versus Et plus Ht. The clarification on token accounting is also helpful.
> >
> > That said, I still have follow-up questions on two core points.
> >
> > 1. On the Evaluate module: the rebuttal clarifies that Evaluate uses a constrained schema and gives an illustrative trajectory example, which is useful. However, my original concern was about validation rather than formatting. I still do not see direct evidence that Evaluate reliably diagnoses genuinely missing information, as opposed to serving as a structured heuristic planning signal. Do the authors have any broader analysis of diagnosis quality, such as agreement with human judgments, reduction in redundant exploration, or success rates of predicted missing items being subsequently resolved?
> > 2. On Algorithm 1: could the authors clarify the semantics of the gap variable more explicitly? In particular, the initialization and stopping condition still seem underspecified to me. A more precise explanation of how g0, gt, and termination interact would make the methodology easier to interpret.

---

> > > ### Author Response · Authors · 2026-04-04
> > >
> > > We sincerely appreciate this follow-up, and we are glad that several of our earlier responses were found helpful. We would like to fully address your remaining concerns below.
> > >
> > > ---
> > >
> > > ### Follow-up 1: Evaluate validation
> > >
> > > > I still do not see direct evidence that Evaluate reliably diagnoses genuinely missing information, as opposed to serving as a structured heuristic planning signal.
> > >
> > > To address this directly, we provide both quantitative statistics and ablation evidence.
> > >
> > > **1. Quantitative statistics on Evaluate behavior.** During development, we observed and debugged agent trajectories and found that most behaviors, including Evaluate, performed as expected. To go beyond these empirical observations, we evaluated Evaluate from two perspectives: (1) whether diagnosed missing items were subsequently resolved, and (2) whether the sufficiency judgments aligned with human assessment. Due to cost and time constraints during the rebuttal period, we randomly sampled 200 trajectories for analysis.
> > >
> > > - **Resolution rate:** Among the 200 sampled trajectories, 61 contained two or more Evaluate actions. We manually inspected all 61. In 54 cases (88.5%), every gap diagnosed by Evaluate was subsequently resolved and committed into $E_t$. The remaining 7 exhibited partially unresolved gaps.
> > > - **Sufficiency judgment accuracy:** We further sampled 50 trajectories (from the 200), which collectively contain 78 Evaluate actions. Of these, 6 produced incorrect early-stopping judgments (diagnosing sufficiency when information was still missing), yielding a **human-aligned accuracy of 92.3%** (72/78). We note that the reverse error (judging insufficient when actually sufficient) is harder to identify manually and is in fact a conservative behavior that the system encourages.
> > >
> > > **2. Ablation evidence.** We also conducted a direct ablation of Evaluate (also reported in our response to Reviewer iGke):
> > >
> > > | Variant | LOOGLE-V2 Acc (%) | ∞BENCH Acc (%) |
> > > |---------|-------------------|----------------|
> > > | SCOUT (Full) | 78.2 | 85.6 |
> > > | w/o Evaluate | 72.7 | 79.3 |
> > > | w/o $E_t$ (ReAct-style) | 70.5 | 75.5 |
> > >
> > > Removing Evaluate leads to a notable accuracy drop on both benchmarks, confirming that it provides meaningful diagnostic signals rather than acting as a generic planning heuristic.
> > >
> > > We believe the combination of quantitative human inspection (88.5% resolution rate, 92.3% judgment accuracy) and macro-level ablation evidence provides reasonable support for the reliability of Evaluate. We will include this analysis in the revised version, and consider a formal alignment study a valuable direction for future work.
> > >
> > >
> > > ### Follow-up 2: Algorithm 1 ($g_t$ semantics)
> > >
> > > > Could the authors clarify the semantics of the gap variable more explicitly? The initialization and stopping condition still seem underspecified.
> > >
> > > Thank you for the follow-up. We are happy to clarify.
> > >
> > > $g_t$ represents the **diagnosed information gap** between $E_t$ and $F^*_q$ (Section 2.3.3). It is produced exclusively by Evaluate, which takes $(q, E_t)$ as input and returns a gap diagnosis that is "empty iff query-sufficient" (Table 5, Appendix B.2).
> > >
> > > - **Initialization:** $g_0 \leftarrow \bot$ (Algorithm 1, line 1). $\bot$ denotes "not yet assessed" and is treated as non-empty, ensuring the loop is entered before any sufficiency judgment.
> > > - **Update:** $g_t$ remains unchanged unless the agent calls Evaluate (lines 6, 12, 19). Only when $a_t$ = Evaluate does $g_{t+1} \leftarrow \text{DiagnoseGap}(q, E_t)$ (line 14).
> > > - **Termination:** The loop exits when $g_t = \emptyset$ (Evaluate finds no remaining gap) or $t = T_{\max}$ (line 2).
> > >
> > > In other words, the stopping criterion is a **state-level sufficiency judgment**. A full notation summary is provided in Table 4 (Appendix A). We will make these semantics more explicit in the revised Algorithm 1.
> > >
> > > ---
> > >
> > > We hope these clarifications address your remaining concern, and we would be happy to discuss further if needed.

---

### Official Review · Reviewer_Ewrb · 2026-03-15

**Soundness:** 4
**Presentation:** 4
**Significance:** 4
**Originality:** 4
**Overall Recommendation:** 5
**Confidence:** 4

**Summary:**

This paper investigates the Trilemma of Long-Text Understanding (LTU), where a system generally struggles to achieve extreme scalability, high information fidelity, and inference efficiency at the same time. To solve this issue, this paper models LTU tasks as a Partially Observable Markov Decision Process (POMDP) and proposes to decouple the procedural trace and epistemic state to isolate the reasoning process from noisy exploration history. The proposed SCOUT method outperforms many state-of-the-art LLMs and Agent systems, with significantly better token efficiency. Further experiments show that the SCOUT framework can also be used to post-train open-weight LLMs to further increase their performance.

**Compliance With Llm Reviewing Policy:**

Affirmed.

**Key Questions For Authors:**

Q1: Could you provide the latency evaluation of SCOUT and other agent / LLM baselines?
Q2: How does the latency scale as the document length increases from 64K to 1M+ tokens compared to the near-constant token cost shown in Figure 3?

**Limitations:**

Yes

**Strengths And Weaknesses:**

Strength:
S1: The paper presents a highly complete and well-structured piece of work. It is grounded in a solid theoretical framework and is validated through extensive experimental evaluation across complex benchmarks.

S2: The proposed SCOUT framework demonstrates substantial performance improvements while significantly reducing token costs. It addresses the fundamental "attention dilution" flaws of the standard ReAct paradigm in long-horizon tasks. This makes SCOUT a highly promising new paradigm for agentic long-text understanding

S3: The framework exhibits impressive scalability. As demonstrated in Figure 3, SCOUT maintains near-constant performance and token cost as context length scales from 64K to 1M+ tokens, successfully alleviating the LTU trilemma in practice.

Weakness:

W1: This paper does not talk about the latency of SCOUT. Better token efficiency does not sufficiently contribute to lower latency, as the serial calls of the system could introduce extra latency which should be discussed.

---

> ### Author Rebuttal · Authors · 2026-03-29
>
> We sincerely thank you for the highly positive assessment and for recognizing the theoretical grounding, experimental completeness, scalability, and paradigm-level contribution of SCOUT. We will respond to your concerns as follows:
>
> ## Latency Evaluation
>
> > **W1:** This paper does not talk about the latency of SCOUT. Better token efficiency does not sufficiently contribute to lower latency, as the serial calls of the system could introduce extra latency which should be discussed.
> > **Q1:** Could you provide the latency evaluation of SCOUT and other agent / LLM baselines?
> > **Q2:** How does the latency scale as the document length increases from 64K to 1M+ tokens compared to the near-constant token cost shown in Figure 3?
>
> ### Token cost vs. latency
>
> We would like to first clarify why the paper focuses on `token cost` rather than latency. Token cost is the primary efficiency metric in this work because it has direct practical implications:
>
> - **API expenditure:** Lower token cost translates to lower monetary cost when using external model services.
> - **Serving capacity:** Under fixed compute budgets, lower token cost per query allows higher concurrency and throughput.
>
> Our primary focus is accuracy, with token cost as a secondary efficiency consideration. Latency optimization is outside the scope of this work. That said, we sincerely appreciate you raising this point, and are happy to provide the latency results below.
>
> ---
>
> ### Latency results
>
> | Method | Category | LOOGLE-V2 Latency (s) |
> |--------|----------|-----------------------|
> | SCOUT | Ours | 153.9 |
> | MemAgent | Agent | 212.4 |
> | Claude-Sonnet-4.5 | LLM | 28.8‡ |
> | Gemini-3-Pro | LLM | 57.2† |
>
> We also measured how latency scales with context length, corresponding to the token-cost scaling in Figure 3:
>
> | Context Length | SCOUT (s) | MemAgent (s) | Gemini-3-Pro† (s) | Claude-Sonnet-4.5‡ (s) |
> |---------------|-----------|-------------|-------------------|------------------------|
> | 64K | 150.0 | 106.3 | 34.5 | 15.1 |
> | 128K | 154.2 | 118.5 | 43.4 | 25.3 |
> | 256K | 154.9 | 177.3 | 58.9 | 36.8‡ |
> | 512K | 157.8 | 310.8 | 84.3 | 36.5‡ |
> | 1M+ | 153.2 | 559.3 | 91.7† | 37.1‡ |
>
> *(Note. As stated in Appendix D.2 (Lines 874-888), these truncations arise from the context window limits of the underlying LLMs: † Gemini-3-Pro is limited to 1M input tokens, and ‡ Claude-Sonnet-4.5 is limited to 200K input tokens.)*
>
> *(Note. For agent methods, latency is the cumulative wall-clock time across all turns (consistent with the token cost definition Line 831-849). For LLM methods, it is the end-to-end response time of a single API call. All measurements were conducted under a unified environment; results should be interpreted as relative comparisons, as absolute values depend on hardware, API provider, and serving configuration.)*
>
> ---
>
> ### Analysis of the Latency Evaluation
>
> - **SCOUT vs. LLMs:** SCOUT has higher absolute latency than monolithic full-context LLMs, which is expected due to multi-turn interaction overhead.
> - **SCOUT vs. Agents:** SCOUT is broadly comparable to MemAgent in overall latency, and exhibits **near-constant latency scaling** from 64K to 1M+, while MemAgent's latency grows significantly with context length.
> - **Positioning:** SCOUT is an **accuracy-first and low-cost** method, not a latency-optimized system. The appropriate choice of method should depend on the **application requirements and deployment scenario, taking into account accuracy, token cost, context length, and latency**. our experiments suggest that SCOUT has clear advantages in: (1) accurate question answering over extremely long texts, (2) low-cost processing of extremely long documents, and (3) raising the LTU performance ceiling of open-source models (Section 3.5).
> - **Concurrency**: Given SCOUT’s substantially lower token consumption (about 1/8 of monolithic LLM-based methods), it can support higher serving concurrency under the same compute budget, partially offsetting its per-query latency disadvantage.
>
> We will include these results in the revised version.
>
> Thank you again for the constructive and encouraging review!

---

> > ### Author Rebuttal · Reviewer_Ewrb · 2026-04-06
> >
> > I will keep positive scores.

---

> > > ### Author Response · Authors · 2026-04-06
> > >
> > > Thank you very much for the positive confirmation! We sincerely appreciate your professional and responsible engagement with our work throughout the review process.

---

### Official Review · Reviewer_gYkA · 2026-03-16

**Soundness:** 2
**Presentation:** 3
**Significance:** 1
**Originality:** 2
**Overall Recommendation:** 3
**Confidence:** 3

**Summary:**

Overall, the paper examines the concept of Long-Text Understanding (LTU) at the million-token scale, addressing the "LTU Trilemma" where current models struggle to balance extreme scalability, high information fidelity, and inference efficiency. This paper focuses on an important concept known as the Information Sparsity Assumption, which posits that only a tiny fraction of a massive document is actually needed to answer any specific query. To exploit this, the authors introduce SCOUT (Strategic Context Observation for Understanding Text), a novel framework that shifts from passively ingesting entire texts or building task-agnostic indexes to actively foraging for information using multi-resolution exploration tools like scanning and targeted reading. Crucially, SCOUT decouples the agent's noisy interaction history from its reasoning process by maintaining a compact, provenance-anchored "Epistemic State" that iteratively gathers only verified, query-relevant facts. By evaluating on complex multi-hop benchmarks like LOOGLE-V2 and ∞BENCH, the researchers demonstrate that SCOUT matches or exceeds the accuracy of frontier models like Gemini-3-Pro and GPT-5.1 while reducing token consumption by up to 8× and maintaining completely stable performance and costs as context windows scale beyond a million tokens.

**Compliance With Llm Reviewing Policy:**

Affirmed.

**Final Justification:**

After reconsideration, I think most of my previously pointed weakness are addressed, except for the following issues:

* I disagree that token cost can completely represent efficiency during deployment. A model that consumes much longer end-to-end latency can affect consumer satisfaction despite using few tokens. In fact, reporting both metrics (latency & token cost) can be a better presentation of the work, so that practitioners can choose the right deployment setting.

* I think the result of Table 9 is not completely a fair comparison without thinking models because the proposed method actually adds more intermediate steps before the answer. This is like comparing a trained thinking model with a trained instruct model.

Although these might not be primary weaknesses, they make the main claim of the paper (active foraging is better than passive reading) unsolified. Thus, I will remain my recommendations.

**Key Questions For Authors:**

1. Could you show me results with thinking mode for section 5.3?
2. Could you show me results of end-to-end latency per question and in batch mode, compared with full-context baseline (with prefix caching turned on so that we reuse kv cache for the same context but different queries)?
3. Could you explain whether the paper is more focused on improving accuracy or efficiency? Let's use SWE environments as an example, how can we improve the accuracy while maintaining low latency?
4. Would you think of your proposed method as a complement to full-context LTU (to solve hard queries) or completely replace full-context LTU?

**Limitations:**

Yes

**Strengths And Weaknesses:**

Overall, the paper explores how to improve long-context understanding via agentic search. While promising results have shown in the paper, there are significant weaknesses as listed below:

**Weakness 1: Unclear Use Case of the Proposed Method**

The proposed method turns long-context understanding into an agentic workflow. Despite the improved accuracy, it is hard to tell whether the end-to-end latency is improved in both streaming and batching scenarios. In fact, I suspect the latency will be significantly larger than the plain long-context because of the multi-turn behavior and the un-sharable KV cache across queries. If the method is completely proposed to improve accuracy for hard queries, then the question becomes how can we trigger the agentic search when the query too difficult to solve with plain long-context. And by-the-way, the paper does measure efficiency, but only in terms of "token cost" instead of "inference latency" which is not the same.

**Weakness 2: Section 3.5 is Unsolid without Thinking Model Experiments**

SCOUT achieves higher accuracy by breaking the reasoning proess into multiple sequential turns. This can be thought of as a way to increase test-time compute. Comparing this multi-step, iterative refinement process against a standard LLM with non-thinking mode will be completely unfair. It remains an open question whether the "active foraging" and synthesis behaviors SCOUT hardcodes would naturally emerge within a native thinking model's reasoning trace if it were simply given the time to "think" over the whole document.

---

> ### Author Rebuttal · Authors · 2026-03-29
>
> Thank you for the detailed feedback! Below we group the related concerns together for clarity.
>
> ---
>
> ### Use Case and Positioning of SCOUT
>
> > **Weakness 1:** Unclear Use Case of the Proposed Method
>
> We have carefully considered your comment and believe that the main concern centers on the **use case and positioning of SCOUT**, including what it optimizes, when it should be used, and how it relates to full-context LTU. We respond point by point below.
>
> > **Q2:** Could you show me results of end-to-end latency per question and in batch mode, compared with full-context baseline (with prefix caching turned on so that we reuse kv cache for the same context but different queries)?
>
> We agree that end-to-end latency is an important systems consideration. To address this question, we additionally measured latency under a unified setup; due to space constraints, **please refer to the latency table in our response to `Reviewer Ewrb`**.
>
> These results show that SCOUT has higher latency than monolithic LLMs but is comparable to agentic baselines, with near-constant latency scaling from 64K to 1M+.
>
> The primary efficiency metric in this paper is `token cost` rather than latency. We would like to clarify that token cost directly affects both **API expenditure when using external model services and serving concurrency in practical local deployments**.
>
> > **Q3:** Could you explain whether the paper is more focused on improving accuracy or efficiency? Let's use SWE environments as an example, how can we improve the accuracy while maintaining low latency?
>
> Our primary concern is **performance (accuracy)**. The targeted design choices, such as state decoupling and evidence anchoring (Section 2.3), are intended to mitigate noise contamination and information loss. As a result, SCOUT exhibits strong scaling behavior, with almost no performance degradation from 64K to 1M+ tokens (Section 3.3). On this basis, our method also achieves good efficiency (`token cost` rather than latency).
>
> We would further like to clarify that both token cost and latency are meaningful metrics. Token cost directly affects processing expenditure and serving cost, which is the aspect this paper is concerned with, whereas latency optimization is outside the scope of the current work.
>
> > **Q4:** Would you think of your proposed method as a complement to full-context LTU (to solve hard queries) or completely replace full-context LTU?
>
> We would like to clarify that SCOUT is complementary to full-context models. In practice, the appropriate choice depends on both the task requirements and the deployment scenario, including *accuracy, cost, context length, and latency.* While frontier full-context LLMs may remain preferable for shorter or simpler inputs, their performance degrades much more rapidly as context length increases (lines 344–347). In that regime, SCOUT becomes the more suitable option.
>
> More specifically, the main advantage of SCOUT is that it **mitigates the LTU trilemma** in practice (Section 3.3, lines 47–54). As you rightly point out, SCOUT may not be the preferred choice when per-query latency is the primary concern. Instead, our experiments suggest that SCOUT is particularly suitable for scenarios such as (1) accurate question answering over extremely long texts, (2) low-cost processing of extremely long documents, and (3) raising the LTU performance ceiling of open-source models (Section 3.5).
>
> We sincerely thank you for this thoughtful question, and we will discuss this point in the revision thoroughly.
>
> ---
>
> ### Thinking Model Experiments
>
> > **Weakness 2:** Section 3.5 is Unsolid without Thinking Model Experiments
> > **Q1:** Could you show me results with thinking mode for section 5.3?
>
> (We respectfully note that Sections 3.5 and 5.3 may be a typo; Section 3.5 studies training gains, and Section 5.3 does not exist in the paper.) We interpret the underlying concern as whether the main results in Section 3.2 should include stronger full-context baselines with thinking capability, and respond accordingly.
>
> We apologize if this point was not sufficiently clear in the paper. In fact, **the paper already includes substantial thinking-model evidence**. On the closed-source side, several baselines in our evaluation are already strong reasoning-oriented models by design, including Gemini-3-Pro, Gemini-2.5-Pro, and the GPT-5 family. On the open-weight side, Appendix E.2 / Table 9 reports full-context results for Qwen3-30B-A3B-Thinking and Qwen3-235B-A22B-Thinking.
>
> **Taken together, these results suggest that simply increasing test-time reasoning does not close the gap on LTU tasks.** One plausible reason is that longer reasoning traces also consume valuable context budget, which can be particularly unfavorable in million-token LTU.
>
> ---
>
> Thank you again for your valuable comments!
>
> We hope this clarification is helpful, and we would be happy to elaborate further if needed.

---

> > ### Author Rebuttal · Reviewer_gYkA · 2026-04-03
> >
> > Thank you for your thoughtful response to my questions. I think I have a clearer understanding now. For the weakness 2, I intended to refer to Section 3.5 (Table 9), as it only reports instruct model performance.

---

> > > ### Author Response · Authors · 2026-04-04
> > >
> > > ### Follow-up: Section 3.5 and thinking models
> > >
> > > > Thank you for your thoughtful response. I intended to refer to Section 3.5 (Table 9), as it only reports instruct model performance.
> > >
> > > Thank you for the clarification. We would like to explain the positioning of this experiment and share our perspective on thinking models for LTU tasks.
> > >
> > > **1. Section 3.5 is a supplementary experiment, independent of the main performance comparison.** Its purpose is to demonstrate that **under matched training budgets, the SCOUT interface provides a better structure for post-training** than monolithic LLM-SFT. We believe the current results already demonstrate this clearly: SCOUT-SFT yields +27.1 / +25.1 gains while LLM-SFT yields only +1.2 / −0.7 (Table 3).
> > >
> > > **2. Why the training-gain gap exists.** The training data characteristics differ fundamentally between the two regimes. As reported in Appendix D.3, LLM-SFT trains on instances averaging ~80K tokens, while the open-weight backbone supports only 128K tokens (extended via YaRN from 32K; Appendix D.2), meaning a substantial portion of training data is subject to middle truncation. Appendix E.4 further discusses why this creates a **length–fidelity tradeoff** that limits LLM-SFT effectiveness. In contrast, SCOUT-SFT trains on trajectories of ~30K tokens (Appendix D.3), where the model learns **exploratory and state-management capabilities** rather than brute-force long-context comprehension. This structural difference is likely the main driver behind the large training-gain gap observed in Table 3.
> > >
> > > **3. On thinking models.** Due to time constraints, we are unable to provide thinking-model results for this setting now, but will include them in the revised version to address your concern. That said, we conjecture that this bottleneck would persist even with thinking models, as the truncation issue stems from the LLM-SFT training regime rather than the model's reasoning capability.
> > >
> > > As Section 3.5 is supplementary and its conclusion is well-supported, we would appreciate it if you could kindly reconsider whether this constitutes a **primary weakness**. We sincerely hope this addresses your concern. Thank you again for your careful and constructive engagement with our work!

---

### Official Review · Reviewer_iGke · 2026-03-18

**Soundness:** 3
**Presentation:** 3
**Significance:** 3
**Originality:** 4
**Overall Recommendation:** 4
**Confidence:** 4

**Summary:**

This paper proposes SCOUT, a framework for understanding long texts that replaces full-context processing with active exploration. The model searches and reads sections of the document in stages, keeping a compact state that contains only relevant information. The main idea is to separate exploration from reasoning and gradually gather enough evidence. Experiments demonstrate better accuracy and cost trade-offs on long-context benchmarks.

**Compliance With Llm Reviewing Policy:**

Affirmed.

**Final Justification:**

The rebuttal addressed most of my concerns and therefore I maintain my positive score.

**Key Questions For Authors:**

None. As seen in weakness

**Limitations:**

yes.

**Strengths And Weaknesses:**

## Strengths

- The paper introduces a clear and intuitive paradigm: treating long documents as an environment to explore rather than fully ingest. This is well motivated for extreme context settings.

- The decoupling between exploration history and reasoning state is a clean design that helps reduce noise accumulation.

- Empirical results on long-context benchmarks are strong, showing good accuracy with significantly lower token usage.

- The framework is conceptually simple and could be extended to other agent-based settings.

## Weaknesses


- The paper only reports token cost but does not evaluate runtime. Since SCOUT relies on multi-step interaction and repeated LLM calls, actual latency may be significantly higher. The current results only support token efficiency, not real system efficiency.


- There is no breakdown of steps, number of model calls, or exploration depth. This makes it hard to understand the true computational behavior and scalability of the method.


- All benchmarks seem long and information-sparse, which aligns with SCOUT’s design. It is unclear how the method performs on more standard tasks (e.g., medium-length QA or summarization), where selective reading may be less effective.


- The exploration strategy assumes relevant evidence can be incrementally discovered. In cases where key information is only indirectly related to the query, the method may miss important evidence. This is not explicitly tested.

- Gap-diagnosis mechanism is not fully validated.
The paper highlights this component as important, but does not provide a direct ablation. Its actual contribution remains unclear.

---

> ### Author Rebuttal · Authors · 2026-03-29
>
> Thank you for your thoughtful and detailed feedback! Your comments are much appreciated. We will respond to your concerns as follows:
>
> > The paper only reports token cost but does not evaluate runtime. Since SCOUT relies on multi-step interaction and repeated LLM calls, actual latency may be significantly higher. The current results only support token efficiency, not real system efficiency.
>
> We appreciate this suggestion. Due to space constraints, please refer to our response to `Reviewer Ewrb` for the full latency evaluation.
> In short, we focus on token cost rather than latency because it directly **reflects API expenditure and serving concurrency**. SCOUT has higher latency than monolithic LLMs but is comparable to agentic baselines, with near-constant latency scaling from 64K to 1M+.
>
> > There is no breakdown of steps, number of model calls, or exploration depth. This makes it hard to understand the true computational behavior and scalability of the method.
>
> We fully agree with your suggestion. To make the computational behavior more explicit, we provide the following aggregate statistics:
>
> | Metric               | LOOGLE-V2 | ∞BENCH |
> |----------------------|-----------|--------|
> | Total steps (mean)   | 31.1      | 27.7   |
> | Total steps (median) | 29.5      | 26.0   |
> | Total steps (min)    | 9         | 7      |
> | Total steps (max)    | 91        | 88     |
> | Avg. Evaluate calls  | 1.8       | 1.5    |
>
> We kindly note that Appendix D.1 and Appendix F already cover multi-turn token cost accounting and a full trajectory example.
>
> > All benchmarks seem long and information-sparse, which aligns with SCOUT’s design. It is unclear how the method performs on more standard tasks (e.g., medium-length QA or summarization), where selective reading may be less effective.
>
> We would like to respectfully clarify that our evaluation setting is not limited to uniformly ultra-long and highly sparse inputs.
>
> First, the benchmarks we use, including LOOGLE-V2 and InfiniteBench, cover a broad range of context lengths rather than only extreme million-token cases. In particular, LOOGLE-V2 spans instances from **16k to over 2M tokens** (Appendix C.2).
>
> Second, our length-binned analysis in **Figure 3 (left)** explicitly evaluates SCOUT across 64K, 128K, 256K, 512K, and 1M+ contexts, and shows that SCOUT maintains consistently strong performance throughout this range. Therefore, the paper already provides evidence that SCOUT performs well not only on ultra-long inputs, but also on shorter and medium-length long-context settings within the benchmark.
>
> > The exploration strategy assumes relevant evidence can be incrementally discovered. In cases where key information is only indirectly related to the query, the method may miss important evidence. This is not explicitly tested.
>
> We would like to clarify that **this case is explicitly tested in our experiments**. LOOGLE-V2 is designed for “true long-context understanding rather than simple retrieval” (Appendix C.2, lines 799-802), and its **Long-Dependency Tasks** require evidence that is “interdependent and scattered throughout the text” and answers that rely on **multi-hop reasoning rather than retrieving a single sentence** (Section 3.1, lines 238-241). Therefore, the benchmark directly evaluates the failure mode raised in the review.
>
> Appendix F provides a concrete qualitative example of exactly this setting: it is **not a retrieval-only task, but a long-dependency multi-hop reasoning task**, where the answer must be composed from constraints distributed across distant sections. In this sense, both our quantitative benchmark and qualitative example already test the kind of indirect evidence discovery concern raised here.
>
> > Gap-diagnosis mechanism is not fully validated. The paper highlights this component as important, but does not provide a direct ablation. Its actual contribution remains unclear.
>
> We agree that a direct ablation would strengthen the paper. The current paper already shows that removing the broader epistemic-state design substantially hurts performance: removing `E_t` drops LOOGLE-V2 from 78.2 to 70.5 (Table 2, lines 1009-1018) and ∞BENCH from 85.6 to 75.5 (Table 10, lines 2577-2585). To directly isolate the contribution of the gap-diagnosis mechanism, we additionally ran an ablation that removes `Evaluate`:
>
> | Variant | LOOGLE-V2 Acc (%) | ∞BENCH Acc (%) |
> |---------|-------------------|----------------|
> | SCOUT (Full) | 78.2 | 85.6 |
> | w/o `Evaluate` | 72.7 | 79.3 |
> | w/o `E_t` | 70.5 | 75.5 |
>
> These results directly confirm that `Evaluate`(Gap-diagnosis mechanism) is an important contributor to SCOUT's gains. We will include these results in the revision to further strengthen the paper.
>
> Thank you again for your valuable comments! We hope our responses have addressed the concerns clearly, and we welcome any further feedback to improve the work.

---

> > ### Author Rebuttal · Reviewer_iGke · 2026-04-01
> >
> > Thank you. My concerns are largely addressed. I will update my confidence accordingly

---

> > > ### Author Response · Authors · 2026-04-01
> > >
> > > Thank you very much for confirming that your concerns have been addressed!
> > >
> > > We sincerely appreciate your time and thoughtful feedback throughout the review process.

---

### Decision · Program_Chairs · 2026-04-30

**Decision:**

Accept (regular)

**Comment:**

This paper proposes SCOUT, a framework for long-text understanding with active exploration over raw text, while maintaining a compact epistemic state for final reasoning. The paper presents a coherent design and shows strong empirical results on challenging long-context benchmarks. My overall judgment is positive: despite some valid concerns about positioning and evaluation scope, I believe the paper makes a valid contribution and achieves the bar for acceptance.

The reviewer feedback was overall favorable. Two reviewers recommended weak accept and one reviewer recommended accept, highlighting the paper’s clear motivation, the value of decoupling exploration history from reasoning state, and the strong empirical results on long-context benchmarks. Reviewers also appreciated the breadth of the evaluation, including comparisons to both long-context LLMs and agentic baselines, along with ablations, scaling analysis, and open-weight training experiments. At the same time, several reviewers raised constructive concerns: the paper focuses on token cost but does not emphasize latency in the main presentation; some claims are phrased too strongly relative to the evidence; and the current evaluation is concentrated on long, information-sparse settings that align well with SCOUT’s design.

A particular point of discussion came from the only slightly negative review, which raised two main concerns. First, it argued that token cost is not the same as deployment efficiency, since latency, batching behavior, and cache reuse also matter in practice. Second, it questioned whether some of SCOUT’s gains may come from the structured multi-step scaffold itself, rather than isolating “active foraging” as the sole source of improvement. I find these concerns valid. The discussion and rebuttal clarified that SCOUT should be viewed as accuracy-first and token-cost-efficient, not latency-optimized, and more as a complement to full-context LTU than a universal replacement. I think this largely resolves the concern at the level of positioning, though the final paper should state these boundaries more explicitly.

Overall, I do not think these issues outweigh the paper’s core merits. The central design is meaningful and likely to be useful beyond this specific benchmark setting. I therefore lean accept. For the final version, I encourage the authors to more clearly distinguish token-cost efficiency from latency efficiency, moderate the stronger claims around “active foraging” and the LTU trilemma, and discuss future evaluation on broader long-context tasks, such as coding-oriented settings beyond the datasets currently used.